# Physics of swimming and its fitness cost determine strategies of bacterial investment in flagellar motility

Irina Lisevich[1], Remy Colin [1], Hao Yuan Yang [1,2], Bin Ni[1,3] & Victor Sourjik [1] ✉

Microorganisms must distribute their limited resources among different physiological functions, including those that do not directly contribute to growth. In this study, we investigate the allocation of resources to flagellar swimming, the most prominent and biosynthetically costly of such cellular functions in bacteria. Although the growth-dependence of flagellar gene expression in peritrichously flagellated *Escherichia coli* is well known, the underlying physiological limitations and regulatory strategies are not fully understood. By characterizing the dependence of motile behavior on the activity of the flagellar regulon, we demonstrate that, beyond a critical number of filaments, the hydrodynamics of propulsion limits the ability of bacteria to increase their swimming by synthesizing additional flagella. In nutrient-rich conditions, *E. coli* apparently maximizes its motility until reaching this limit, while avoiding the excessive cost of flagella production. Conversely, during carbon-limited growth motility remains below maximal levels and inversely correlates with the growth rate. The physics of swimming may further explain the selection for bimodal resource allocation in motility at low average expression levels. Notwithstanding strain-specific variation, the expression of flagellar genes in all tested natural isolates of *E. coli* also falls within the same range defined by the physical limitations on swimming and its biosynthetic cost.

Microorganisms, like all living systems, must achieve multiple physiological objectives, which may change when they encounter new environments. They have therefore evolved numerous regulatory mechanisms responsible for distributing their limited resources to specific physiological functions[1,2]. Bacteria, including *Escherichia coli*, have become a convenient model to address this fundamental resource allocation problem[3], with a primary focus on proteome partitioning[4–7]. To allocate their resources into protein biosynthesis as a function of growth rate, bacteria appear to obey linear rules known as growth laws[4,5]: the fraction of the proteome responsible for biomass production expands with growth rate, whereas the fraction

responsible for nutrient uptake and catabolism decreases with growth rate. This leads to the negative linear correlation between the expression of carbon catabolic genes and growth rate that is known as the C-line[5], which has been proposed to maximize growth. However, although growth maximization is an important strategy[4–6,8], it does not always apply[9,10] and cells may instead prioritize other targets such as energy yield or stress response[11,12]. Furthermore, while previous studies have mostly focused on the optimized expression of catabolic[4,5,8,10], anabolic[5,6] or ribosomal[2,5,6] genes, how microbes allocate resources to multiple functions that are not directly required for growth remains unclear[13].

[1]Max Planck Institute for Terrestrial Microbiology & Center for Synthetic Microbiology (SYNMIKRO), Karl-von-Frisch-Strasse 14, Marburg, Germany. [2]Max Planck School Matter to Life, Jahnstraße 29, Heidelberg, Germany. [3]College of Resources and Environmental Science, National Academy of Agriculture Green Development, China Agricultural University, Yuanmingyuan Xilu No. 2, Beijing, China. ✉e-mail: victor.sourjik@mpi-marburg.mpg.de

The most prominent example of such a costly physiological function is swimming motility. Motile bacteria are propelled by the rotation of long helical flagellar filaments powered by a motor that is typically proton-driven[14], which enables them to follow spatial gradients of nutrients or harmful chemicals sensed by the chemotaxis signaling pathway[15,16]. The biosynthesis of the motility apparatus (primarily flagellar filaments) consumes several percent of the total cellular protein budget in *E. coli* and other bacteria (2–8% in *E. coli*, depending on growth conditions)[2,17–20], imposing a major burden on cell growth. Consistent with this high fitness cost, a clear trade-off between growth and motility was observed during experimental evolution of *E. coli*[21–23].

The flagellar regulon in *E. coli* is controlled by catabolite repression[24], such that flagellar gene expression increases in minimal medium during growth on poor carbon sources in accordance with the C-line[7,25]. The physiological relevance of such an investment strategy remains debated. One proposed explanation is that it ensures an anticipatory allocation of resources towards motility, in proportion to the potential benefit of finding additional nutrient sources via chemotaxis, which is higher in nutrient-poor environments[25]. Another, related, interpretation is that an increased investment in motility in nutrient-replete conditions may promote the chemotaxis-driven range expansion of bacterial populations[26]. Alternatively, it has been suggested that an elevated flagellar gene expression at lower growth rates compensates for changes in the bacterial cell size, in order to maintain a constant minimal number of flagella per cell that is needed to keep most of the population motile[27]. While these interpretations are not necessarily mutually exclusive, they may also be affected by the physiology of the particular *E. coli* strain or by the growth conditions being used. Moreover, while these studies primarily focused on explaining the relative growth-dependent changes in flagellar gene expression, the dependence of motility on the absolute investment was only explored within the range below the native level of flagellar gene expression[27]. In contrast, the factors that determine the upper physiological limit of resource allocation in motility remained unknown.

In this study, we aimed to more comprehensively understand the motility-growth trade-off over a wider range of flagellar expression levels, quantifying the relation between the expression of flagellar genes and motile behavior, as well as the impact of motility on the growth fitness of *E. coli*. We further compared strategies of resource investment used by a number of laboratory strains and natural isolates of *E. coli*. We demonstrate that major limitations on resource investment in motility, at both high and low levels of gene expression, arise from hydrodynamic constraints on bacterial swimming. Although bacterial motility initially increases with the investment in expression of flagellar genes, consistent with the previous study[27], it saturates at high levels of expression despite further elevation in the number of flagella. Mathematical modelling suggests that this saturation is due to the physical limitation on *E. coli*'s ability to enhance its swimming by adding more than a certain number of flagella. Together with the fitness cost of flagellar biosynthesis and operation, this creates the physiologically relevant range within which the expression level of flagellar genes can enhance motility at a cost of growth fitness.

We observe that within this range, strategies of resource allocation towards motility depend on the medium, growth rate and *E. coli* isolate. In the nutrient-rich medium, *E. coli* MG1655 and other tested K-12 strains appear to maximize their motility up to the physical limit while avoiding an excessive cost of expression beyond this limit. Motility-growth trade-off is also observed during carbon-limited growth, where motility is however not maximized, but rather varies inversely proportionally to the growth rate and appears to correlate with the level of motility at which the benefit of chemotaxis towards additional nutrients saturates. This finding is consistent with the previous report for *E. coli* MG1655[25] but contrasts with the conclusions drawn for another *E. coli* K-12 strain HE204[27]. Furthermore, the

expression of flagellar genes becomes bimodal at very low expression levels, possibly to avoid the production of poorly motile cells. Finally, the motility of natural *E. coli* isolates is limited to the same range as observed for the K-12 derivatives, although its levels vary between isolates and, in some cases, are sensitive to the mechanical properties of the growth environment, probably reflecting the niche-specific selection.

## Results

### Native regulation of flagellar genes in nutrient-rich medium maximizes swimming while limiting the cost of expression

To investigate how motility and growth depend on the allocation of cellular resource to flagellar biosynthesis over a broad range of expression levels, we engineered a derivative of *E. coli* K-12 strain MG1655 with titratable expression of the *flhDC* operon that encodes the master activator of the entire flagellar regulon (Fig. 1a, Supplementary Data 1 and Methods). Different from a recent study with a similar design based on the $P_{tet}$ promoter[27], we utilized a stronger $P_{tac}$ promoter inducible by isopropyl β-D-1-thiogalactopyranoside (IPTG). To reduce its basal activity, the *lacI* repressor gene was inserted upstream of the $P_{tac}$ promoter and the strain was transformed with pTrc99a plasmid vector carrying another copy of *lacI*. For comparability and to be consistent with the previous work[25], other strains were also transformed with pTrc99a. Expression of the flagellar regulon in the resulting *Ptac* strain at different levels of $P_{tac}$-*flhDC* induction was quantified using a green fluorescent protein (GFP) reporter for flagellin (*fliC* gene) promoter activity ($P_{fliC}$), which was previously shown to well reflect the production of flagella in *E. coli*[22,25,28]. Reporter activity was measured using either a plate reader to follow changes in the mean expression over time (Supplementary Fig. 1a, b), or flow cytometry to determine the distribution of single-cell expression levels within the cell population at a defined time point in mid-exponential phase (Fig. 1b). We confirmed that both readouts yielded similar results for *E. coli* cultures grown in nutrient-rich tryptone broth (TB) medium, and that our design of the inducible *Ptac* strain enabled covering the expression levels of flagellar genes both well below and well above the native MG1655 (wild-type; MG1655 *WT*) expression (Fig. 1c and Supplementary Fig. 1b). This is in contrast to the previous study[27], which only covered the expression range below the native level in strain HE204 that was engineered to have an MG1655-like regulation of flagellar gene expression[26].

To investigate how motility changes as a function of gene expression below and above the wild-type levels, we characterized swimming behavior in populations of MG1655 *WT* and *Ptac* cells grown in a batch culture for 4-5 generations to the mid-log phase ($OD_{600} = 0.4$–0.6) using differential dynamic microscopy (DDM)[29] (see Supplementary Note 1 and Supplementary Fig. 2). We observed that population-averaged cell swimming velocity initially increased with expression at low levels of induction, but saturated at high levels of expression (Fig. 1d). Notably, this saturation occurred around the level of flagellar gene expression seen in the wild-type strain. A similar pattern was observed when either the fraction of well-swimming cells within the population, as determined by our motility assay, or the swimming velocity of only these cells were plotted individually (Supplementary Fig. 1c, d). The cell swimming velocity at the highest expression level was even slightly reduced (Fig. 1d and Supplementary Fig. 1d). Notably, although flagellar gene expression and motility are well known to be growth-phase dependent in wild-type *E. coli* strains[30,31] (Supplementary Fig. 1b), no apparent effect of sampling $OD_{600}$ was observed in the narrow range used in our experiments (Supplementary Fig. 1e).

Two other derivatives of *E. coli* K-12, W3110[32] and RP437[33] that are both wild-type for motility, mapped to the same part of the expression-motility curve as MG1655 *WT* and were only slightly less motile (Fig. 1d), due to the lower flaction of motile cells for W3110 and to the

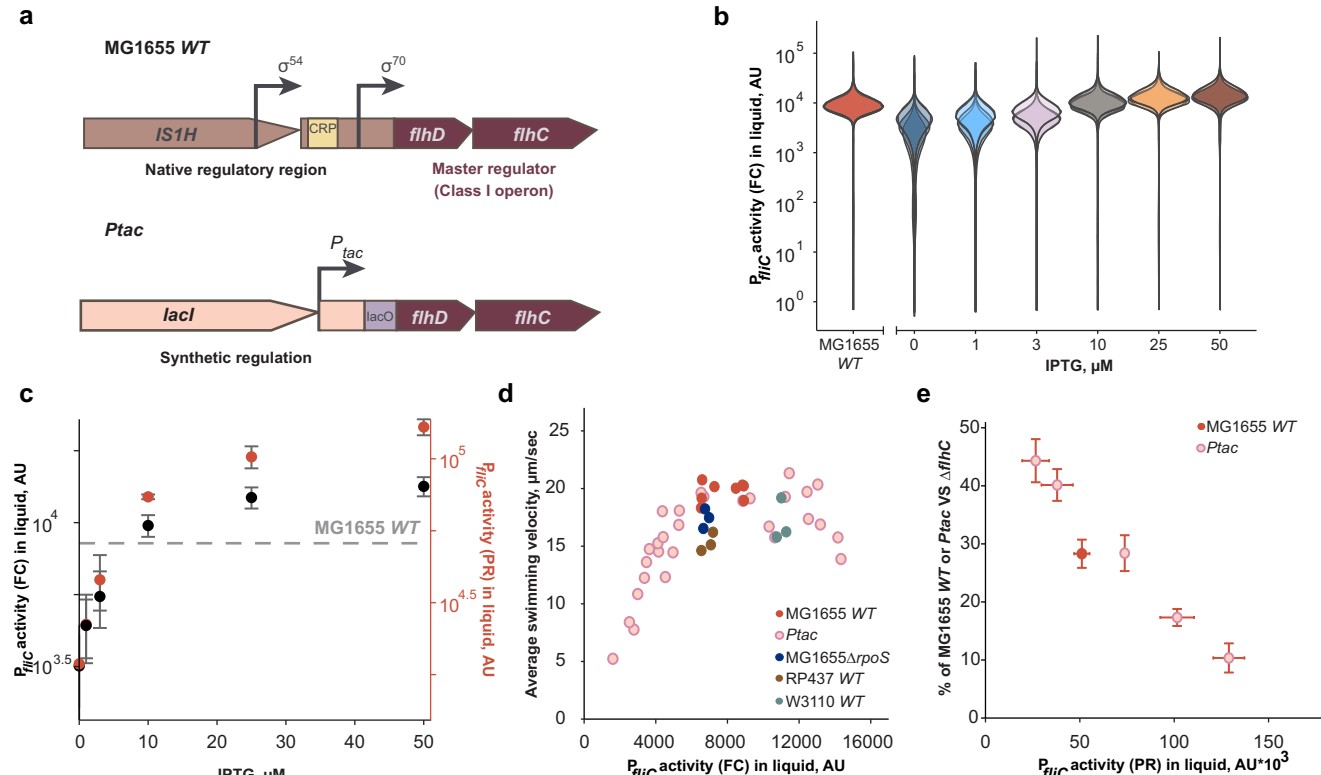

**Fig. 1 | Dependence of motility and its cost on the expression of flagellar genes in nutrient-rich medium. a** Schematic representation of the *flhDC* operon in *E. coli* K-12 strain MG1655, with native (MG1655 *WT*) or inducible (*Ptac*) regulation of expression. The native regulatory region of the *flhDC* operon, including the upstream *IS1H* insertion element, was replaced in the *Ptac* strain with the *tac* promoter inducible by isopropyl β-D-1-thiogalactopyranoside (IPTG) and a copy of the *lacI* gene (Lac repressor) was inserted upstream of the *tac* promoter to reduce the basal expression. As done previously[25], the expression was further reduced in *Ptac* strain by transforming the pTrc99a plasmid vector carrying *lacI*; for a proper comparison, all K-12 strains also carried pTrc99a. **b** Flow cytometry (FC) measurement of P*fliC*-GFP reporter activity in mid-exponential cultures of MG1655 *WT* (IL28, see Supplementary Data 1) or its *Ptac* derivative (IL29) grown in tryptone broth (TB) medium. Data are for three biological replicates (*n* = 3) shown in different hues (AU – arbitrary units). Flagellar gene expression in the *Ptac* strain was induced with the indicated concentrations of IPTG. **c**, P*fliC* reporter activity determined either as the median GFP intensity at mid-exponential growth phase in flow cytometry measurements (black symbols) or as the peak of GFP expression normalized by $OD_{600}$ in plate reader (PR) cultures

(red symbols). Both data sets were aligned by MG1655 *WT* expression (horizontal dashed line). Points are the mean values (*n* = 3) and error bars are the standard deviations (mean ± s.d.). **d** Dependence of the population-averaged cell swimming velocity, calculated as the product of the swimming fraction and the swimming velocity of motile cells, in cultures of the indicated *E. coli* strains (IL28, IL29, IL121, IL149, and IL146), on the activity of the P*fliC* reporter as determined by flow cytometry. Motility and reporter expression were measured separately for each replicate culture (indicated by individual symbols). **e** The growth fitness cost of flagellar gene expression. Fitness cost was determined as the percentage of cells (in %) of either the MG1655 *WT* or *Ptac* strain (labeled with CFP; IL26 or IL107, respectively), induced by different concentrations of IPTG in co-cultures with the non-flagellated *ΔflhC* strain (labeled with YFP; IL25) after 24 h of growth with shaking (200 rpm) in TB medium. The strains were initially co-inoculated in a 1:1 ratio. P*fliC* activity measured in the plate reader was used to plot the data; shown are the mean values (*n* = 3; mean ± s.d.) of P*fliC* activity and of fitness cost measured in three independent cultures of individual strains or in co-cultures with *ΔflhC*, respectively. Source data are provided as a Source Data file.

lower swimming speed for RP437 (Supplementary Fig. 1c, d). Although RP437 is commonly studied as a wild type for *E. coli* chemotaxis, its poorer swimming performance may be a consequence of its extensive mutagenization[33]. Consistently, a previous study showed that the motility of this strain can be improved by experimental evolution[22]. Furthermore, since activity of flagellar regulon is counterregulated by the stationary phase sigma factor RpoS ($\sigma^S$)[34–36], we also measured the impact of the *ΔrpoS* deletion on the expression-motility relation. Both flagellar gene expression and motility of MG1655*ΔrpoS* were similar to those in the wild-type MG1655 (Fig. 1d and Supplementary Fig. 1c, d), suggesting that the well-known *rpoS* polymorphism among *E. coli* strains[37] is unlikely to affect the observed dependence of motility on gene expression in the mid-log phase.

Since much of the previous work on the cellular resource allocation in *E. coli*, including investment motility, has been performed under the conditions of balanced growth[1,26,27,38], we also measured the dependence of motility on gene expression during balanced growth for MG1655 *WT* and for the lowest and highest induction levels in *Ptac*

strain. These balanced growth data mapped well to the expression-swimming curve obtained under batch culture conditions (Supplementary Fig. 1f). Most importantly, flagellar gene expression and motility in MG1655 *WT* mapped to the same point where motility plateaus, confirming that our conclusions also hold under balanced growth. Furthermore, the overall dependence of motility on expression and the position of MG1655 *WT* in these experiments were not affected by the absence of the *lacI*-carrying pTrc99a plasmid (Supplementary Fig. 1f), although the basal expression in *Ptac* strain increased as expected.

We further investigated the cost of motility in a competitive growth assay, as commonly done to quantify relative growth fitness of bacteria[39], by co-culturing CFP-labeled MG1655 *WT* or *Ptac* strains with a non-flagellated YFP-labeled *ΔflhC* strain. The fitness cost of flagellar regulon activity over a culture passage was determined as the reduction in relative cell number of the tested strain in the co-culture from the initial 50% at inoculation, as done previously[22,25]. This cumulative fitness cost gradually increased with the level of flagellar genes

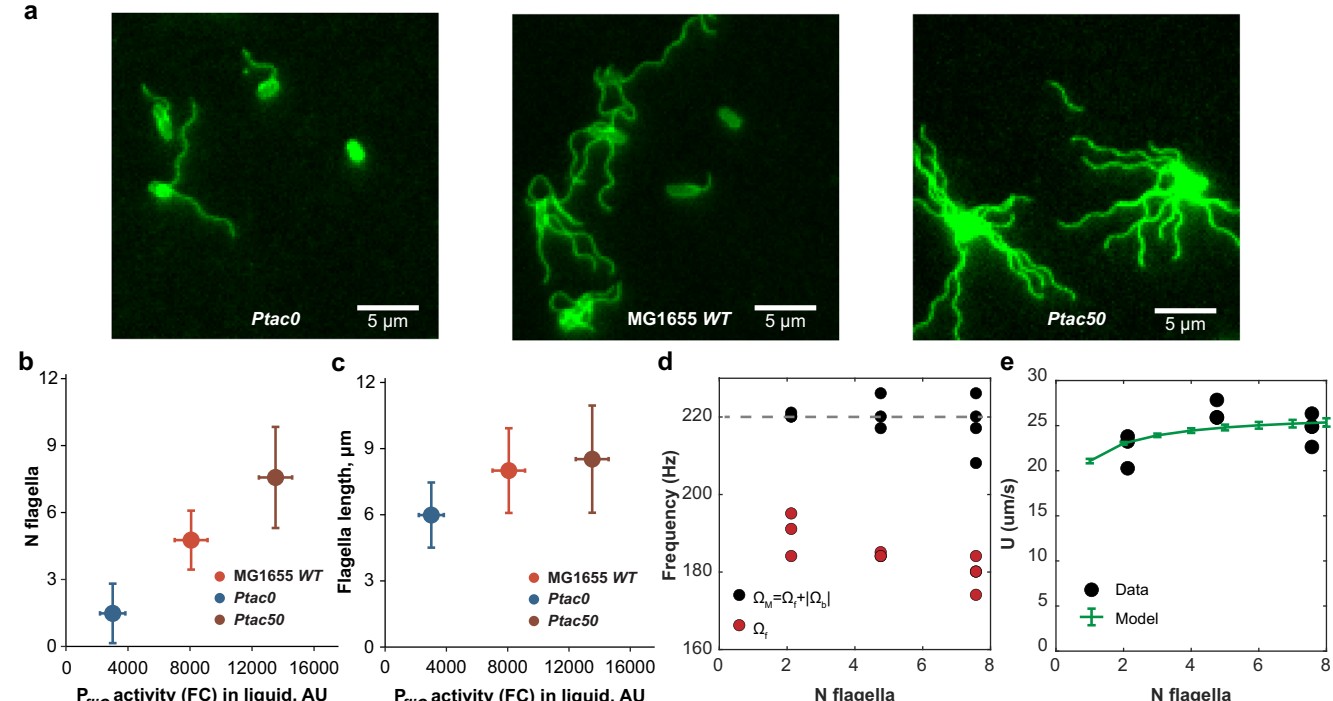

**Fig. 2 | Limitation of *E. coli* motility at high expression of flagellar genes.**
**a–c** Changes in *E. coli* flagellation with varying expression of flagellar genes. Fluorescence microscopy images of MG1655 *WT* (IL28) or *Ptac* (IL29) cells grown either without (*Ptac0*) or with 50 μM IPTG (*Ptac50*), stained with amino-specific fluorescent dye to visualize flagella (**a**). Corresponding quantification of the number (**b**, N flagella) and length (**c**, in μm) of flagella as a function of $P_{fliC}$ activity measured by flow cytometry (FC, $n = 3$ biological replicates, mean ± s.d.). The same experimental cultures were used to quantify both the number and length of flagella; $n = 106$ cells from different fields of view (**b**) and $n = 35, 46, 47$ flagellar filaments in 10, 20 and 7 cells of MG1655 *WT* (IL28), *Ptac0* and *Ptac50* (IL29), respectively (**c**). The analysis of flagellar lengths was done using different numbers of cells to ensure that the number of filaments is comparable between conditions (see also Supplementary Fig. 3 for value distributions and significance analysis). **d** Measured rotation frequencies. The black dots are motor frequencies (sum of flagellum and cell body rotation rates), and the red dots are flagellar rotation frequencies, measured as illustrated in Supplementary Fig. 2. The gray dotted line is the motor frequency used in the model (220 Hz). **e** Swimming velocity as a function of the number of flagellar filaments for MG1655 *WT* (IL28), *Ptac0* and *Ptac50* (IL29) strains, as well as the prediction by the physical model of the multi-flagellated microswimmer. Each dot represents an independent biological replicate. The model takes into account that cells with a higher number of flagella also have longer filaments, as observed experimentally. The error bars represent the standard deviation over $n = 5000$ modeled cells with cell body size and motor orientations drawn randomly from expected distributions (see Supplementary Note 2 for details). The number of flagellar filaments in (**d**, **e**) is the average over only flagellated cells for the given strain. Source data are provided as a Source Data file.

expression over the entire range of induction tested (Fig. 1e). Thus, expression of flagellar genes beyond the native level in *E. coli* K-12 strains does not appear to provide any additional benefit, but nevertheless imposes an increasing fitness cost.

## Hydrodynamic constraints limit cell velocity at high levels of flagellar production

The saturation of *E. coli* motility at high levels of flagellar gene expression could be due either to some bottleneck in the biogenesis of functional flagella or to limits in the physical propulsion by multiple flagella. To distinguish between these two possibilities, we first determined how the activity of the flagellar regulon corresponds to changes in flagellation. Staining flagella with an amino-specific fluorescent dye[40] revealed a clear dependence of the number and length of flagella on the expression of the flagellar regulon (Fig. 2a). The average number of flagellar filaments per cell showed an approximately linear increase with the activity of the $P_{fliC}$ reporter (Fig. 2b, Supplementary Fig. 3a). The length of flagellar filaments also showed a moderate increase followed by an apparent saturation (Fig. 2c, Supplementary Fig. 3b). These results were consistent with increased amounts of intra- and extracellular flagellin, determined by immunoblotting (Supplementary Fig. 4). Importantly, this correlation was observed not only below the wild-type expression levels, as reported before[27], but also well above those in MG1655 *WT*. Thus, *E. coli* cells can synthesize more flagella at levels of flagellar gene expression that exceed those of wild-

type cells, but this increase does not translate into higher swimming velocity.

Alternatively, this saturation of swimming with flagellar number could be explained by the physics of *E. coli* motility. The hydrodynamics of flagella-propelled bacterial swimming can be captured by mathematical models[41,42] that consider the balance between the hydrodynamic friction forces and torques on the flagellum, obtained from resistive force theory (RFT)[43,44], and the cell body[45,46]. We therefore used this force balance analysis to describe the swimming of a multi-flagellated bacterium (Supplementary Note 2 and Supplementary Fig. 5). We assume that multiple flagella form a tight bundle that rotates to propel the cell, with the experimentally determined increase of flagellar length and flagellar number as a function of gene expression (Fig. 2a–c). The model also accounts for the elongated shape of the bacterial cell body and its wobbling around the direction of motion, and for flagellar motors being evenly distributed along the cell body (Supplementary Fig. 5a–c). Furthermore, our experimental measurements of flagellar and cell body rotation frequencies in swimming cells suggest that motors operate at a constant speed of approximately 220 Hz that does not depend on the number of flagella (Fig. 2d).

Although existing versions of the RFT models[43,44] use different expressions for the flagellar friction coefficients, leading to quantitative differences in their predictions, the fractional change of velocity of swimming cells and flagellar and cell body rotation frequencies with

the number of flagella is well predicted by all models (Supplementary Fig. 5d–f), and experimental data could be quantitatively matched by adjusting a single parameter in the model (Fig. 2e). The initial increase stems from the increase of flagellar length and the increased thickness of the bundle formed by multiple flagella. Saturation then occurs in the RFT model at high number of filaments because the viscous drag of the cell body becomes negligible compared to the drag of the flagella themselves. As a consequence, any increase in thrust resulting from adding more flagella is offset by an equal increase in viscous drag, since the two have identical dependencies on flagellar length and bundle thickness. Similar effects are at play upon increased flagellar length in monoflagellated bacteria[42]. The model might overestimate the motility of *E. coli* with a single flagellum, because its assumption of a rigid filament is valid for flagellar bundle but not necessarily for an individual flagellum. The model also tends to underestimate the cell wobbling angle $\theta$, likely because it neglects the hook elasticity[47], but the predicted dependence of the swimming speed on flagellar number changes little when using more realistic values of $\theta$[48] (Supplementary Fig. 5g, h).

The constant motor speed further implies in our model that the load per motor is low even for single flagellated cells and decreases as the number of flagella increases (Supplementary Fig. 5i), because multiple motors share the torque generation necessary for bundle rotation and cell propulsion. Although the model neglects possible additional torque resulting from solid friction between flagella, this result implies that under our conditions the motors operate in the low-torque regime at near-maximum speed, reported to vary from 100 to 300 Hz dependent on temperature[49]. This interpretation would also be consistent with other experimental observations[50–52]. Alternatively, our results could indicate the existence of some hypothetical compensatory regulation of the motor speed dependent on load below its maximum (see Supplementary Note 2 for discussion).

Summarily, although our model is clearly simplified and does not capture all the complexity of flagella bundle hydrodynamics[53], it strongly indicates that the ability of *E. coli* to increase its swimming velocity by increasing the number and length of flagella is indeed limited by the hydrodynamics and mechanics of flagellar propulsion in viscous media.

### Flagellar gene expression under carbon-limited growth does not maximize motility

We next investigated whether the C-line-dependent regulation of *E. coli* flagellar genes in carbon sources that support different growth rates[25] similarly serves to maximize swimming, or to maintain a constant level of motility as suggested before[27], or whether it may optimize an alternative target. Consistent with previous studies[7,24,25], the expression of flagellar genes in the MG1655 *WT* strain grown in the minimal medium was much lower in the presence of a good (glucose) than a poor (succinate) carbon source (Fig. 3a). Expression in the *Ptac* strain at a given induction was also lower during growth on glucose, but this dependence was weaker, as expected for promoters that are not catabolite repressed[54]. Importantly, the average swimming velocity (Fig. 3b) as well as growth fitness cost (Supplementary Fig. 6a) in the *Ptac* strain showed the same dependence on flagellar gene expression for both carbon sources in the minimal medium. In the previous study[27], motility of cells grown on different carbon sources only mapped to the same dependence when expression was plotted in units per cell, rather than per biomass, which is likely consistent with our flow-cytometry based measurements that rather reflect the total amount of fluorescent reporter protein per cell.

Different from the growth in nutrient-rich medium, the swimming velocity of MG1655 *WT* cells growing in minimal medium was clearly not maximized and depended strongly on the carbon source (Fig. 3b). Similar result was obtained under balanced exponential growth and without pTrc99a (Fig. 3b and Supplementary Fig. 6b), which is closest

to the experimental conditions used by Honda et al.[27]. Thus, the much weaker dependence of motility on the growth rate reported in that study is most likely due to the difference between *E. coli* strains (see Discussion).

Since in minimal media MG1655 *WT* did not maximize swimming, the native gene expression under carbon-limited growth might correlate with the potential benefit that could be achieved by performing chemotaxis towards sources of additional nutrients. Indeed, not only cell motility but also the benefit of chemotaxis in the presence of nutrient gradients increase in poor carbon sources[25]. Following this previous study, we measured the benefit of chemotaxis by providing localized sources of amino acids in co-culture between the *Ptac* strain (labeled with CFP) and its motile but non-chemotactic Δ*cheY* derivative (labeled with YFP) for different levels of flagellar gene induction (Supplementary Fig. 7a). Besides confirming previous finding that such measured fitness benefit of chemotaxis was generally higher in the presence of succinate as the primary carbon source compared to glucose[25], we observed that also the saturation of chemotaxis benefit occurred at higher expression levels of flagellar genes in succinate (Supplementary Fig. 7b). Importantly, the point of saturation was close to the native level of expression in the respective carbon source.

### Bimodality of flagellar regulon activity at low expression levels

Another notable finding was the appearance of two distinct subpopulations, with almost negative and strongly positive expression, at low average levels of reporter activity in the *Ptac* strain (Fig. 3a and Supplementary Fig. 8). Interestingly, this separation is a function of the average reporter activity and does not depend on the carbon source (Fig. 3a, c), and it was also observed at low expression in TB (Supplementary Fig. 9a, b). In this low expression range, the proportion of positive cells in the population increased up to a critical level of expression, after which the distribution became unimodal and it was rather the mean of the positive peak that increased with expression (Fig. 3c).

Flagellar gene expression in MG1655 *WT* cells grown under our standard conditions was above the critical level where bimodal behavior becomes apparent, even in the presence of glucose. To determine whether bimodality may also emerge in the wild type, we further reduced flagellar gene expression using prolonged growth under catabolite repression in glucose, either by starting from a higher (1:1000) dilution of the TB-grown overnight culture or pre-growing the overnight culture in glucose (Fig. 3c, Supplementary Fig. 9c). Indeed, upon such reduction the MG1655 *WT* cell population exhibited a bimodal pattern of expression similar to that observed in the *Ptac* strain. Thus, bimodality appears to depend solely on the expression level and not on the details of transcriptional regulation of the *flhDC* operon, on the growth medium or on the dilution factor.

Flagellar gene expression has previously been shown to be pulsatile in *E. coli*[28,55] and bistable in the closely related species *Salmonella enterica*[56], and both bistability (in *S. enterica*) and pulsatility (in *E. coli*) of expression were attributed to negative regulation of FlhDC activity by YdiV (RflP)[55,57]. We therefore tested whether regulation by YdiV could be responsible for the emergence of bimodality in our experiments. As expected, the expression level of flagellar genes in MG1655Δ*ydiV* strain was elevated, and it was above the bimodality threshold in glucose even when the culture was inoculated from TB at a 1:1000 dilution (Fig. 3c and Supplementary Fig. 9c). However, the batch culture inoculated from the overnight pre-culture grown in glucose showed the expression level that was sufficiently lowered, and two distinct subpopulations could be clearly distinguished in both the MG1655Δ*ydiV* and *Ptac*Δ*ydiV* strains (Supplementary Fig. 9c, d and Fig. 3c). These data suggest that negative regulation by YdiV is not sufficient to explain the bimodal activation of the P*fliC* reporter. We next investigated the involvement of another negative regulator, the

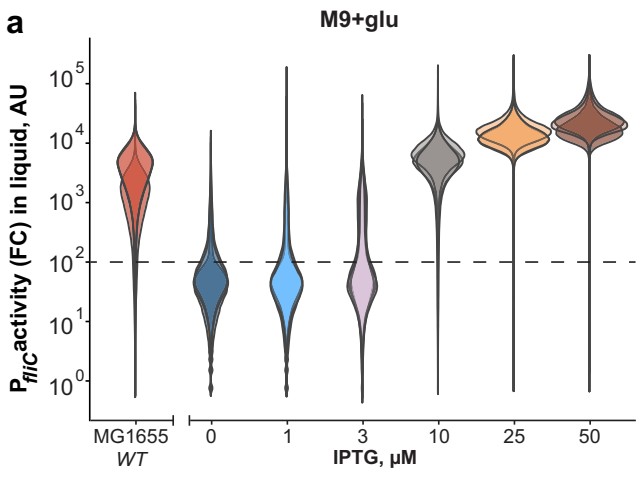

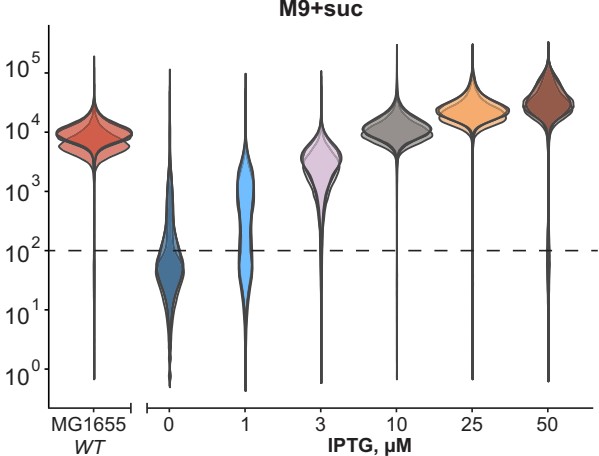

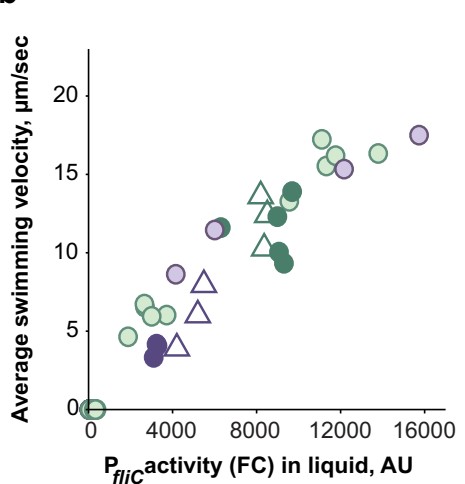

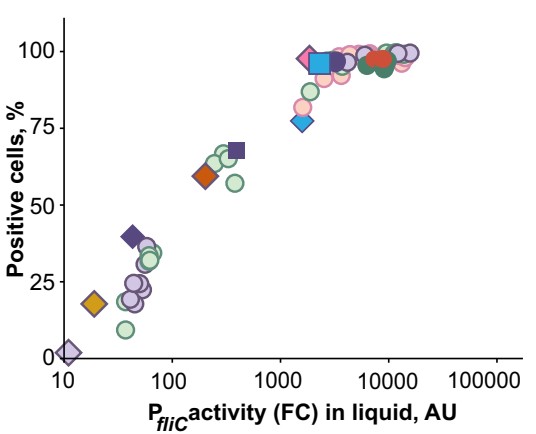

**Fig. 3 | Motility of *E. coli* as a function of gene expression in minimal medium.**
**a** Flow cytometry measurements of the P*fliC*-GFP reporter of MG1655 *WT* (IL28) or *Ptac* (IL29) strains grown to mid-exponential phase in M9 minimal medium, with either glucose (left) or succinate (right) as the sole carbon source. Labels for different induction levels of *Ptac* are the same as in Fig. 1b. Flow cytometry histograms of three biological replicates are shown as violin plots in different hues (AU – arbitrary units). Horizontal dashed line indicates threshold P*fliC* activity level for cellular auto-fluorescence defined as the signal from cells without fluorescent reporter (see also Supplementary Fig. 8). **b** Dependence of the population-averaged swimming velocity on the median P*fliC* reporter activity (flow cytometry, FC) for the carbon sources and strains indicated in the table below. Each dot

represents an independent culture (biological replicate) for which both expression (P*fliC* reporter activity) and swimming were determined. Balanced growth experiments were performed with MG1655 *WT* not carrying pTrc99a (IL182). Flagellar gene expression in *Ptac* strain was induced as in panel (**a**), with 0 to 25 μM IPTG in glucose and 0 to 10 μM IPTG in succinate. **c** Percentage of GFP-positive cells within the population of the indicated *E. coli* strains (IL28, IL29, IL164, IL165, IL175, and IL217) as a function of median P*fliC* reporter activity, both measured by flow cytometry as in (**a**). Each symbol represents an independent culture, with strains and growth conditions indicated in table below. Data are from the batch growth experiments shown in Fig. 1d, (**b**), Supplementary Fig. 1f, Supplementary Fig. 6b, and Supplementary Fig. 9. Source data are provided as a Source Data file.

anti-sigma factor FlgM that inhibits the expression of class 3 flagellar genes including *fliC*[58]. The deletion of *flgM* in *Ptac* strain expectedly increased the average P*fliC* activity, and it resulted in a broad and only weakly bimodal distribution of expression levels (Supplementary Fig. 9e). Bimodality could no longer be seen in *PtacΔflgMΔydiV*

(Supplementary Fig. 9f), but the average expression in this strain remained high and could not be reduced below the bimodality threshold. Thus, negative regulation mediated by both YdiV and FlgM is important for the bimodal expression of flagellar genes, with FlgM apparently playing a more important role.

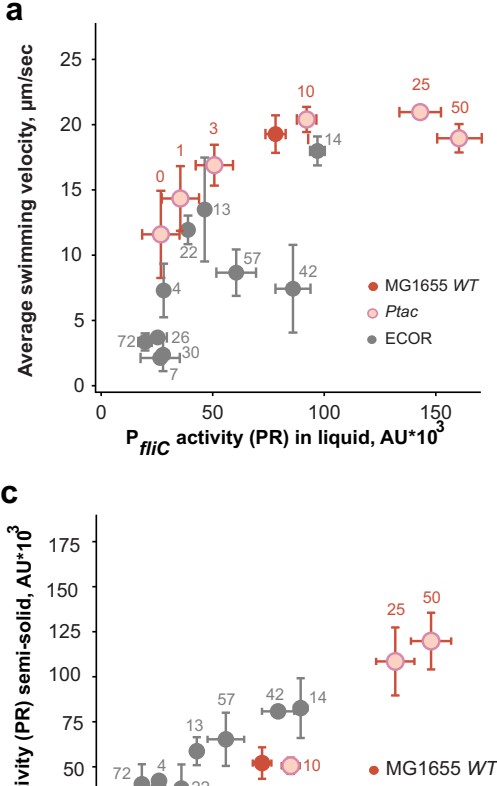

**Fig. 4 | Activity of the flagellar regulon and motility in natural isolates of *E. coli*.** **a** Relation between flagellar regulon activity and motility for representative ECOR strains (indicated here and throughout by their number in the collection) compared to MG1655 *WT* (IL28) and *Ptac* (IL29) strains grown in a liquid TB medium; corresponding inducer concentrations (IPTG, μM) used for the *Ptac* strain are indicated by numbers in red. The same mid-exponential cell culture was used to measure the $P_{fliC}$ reporter activity in the plate reader (GFP fluorescence normalized to $OD_{600}$) and a population-averaged swimming velocity (see Methods for details). Each point represents the mean value for both parameters measured in three independent cultures of each strain ($n = 3$), with error bars indicating the standard deviations. **b** Diameters of spreading zones formed by MG1655 *WT*, *Ptac* and ECOR strains in porous 0.27% TB agar, measured after 4–5 h incubation at 34 °C Bar charts represent the mean diameter across three individual colonies of each strain (overlaid as empty black circles) and error bars indicate standard deviations. **c** Correlation between $P_{fliC}$ reporter activity of in *E. coli* strains grown in liquid or on the semi-solid medium (0.5% TB agar) ($n = 3$ biological replicates; mean ± s.d.). **d** Dependence of the population-averaged swimming velocity on $P_{fliC}$ activity for ECOR, MG1655 *WT* and *Ptac* strains grown on semi-solid medium ($n = 3$ biological replicates, mean ± s.d.). Data for other ECOR strains are shown in Supplementary Fig. 10. Source data are provided as a Source Data file.

## Activity of the flagellar regulon and motility in natural isolates of *E. coli*

Finally, to investigate how investment in motility varies among *E. coli* strains that may have adapted to different ecological niches, we used the ECOR collection, which contains 72 isolates from different hosts and geographical regions[59]. From this collection, we first selected 61 strains that were sensitive to kanamycin and thus transformable with the $P_{fliC}$ reporter plasmid, and then discarded 23 non-swimming isolates that did not spread in porous (0.27%) TB agar. From the remaining 38 spreading isolates, a subset of 24 strains with moderate and good spreading abilities was chosen for further investigation (Supplementary Data 2). Since the presence of pTrc99a had no impact on either cell motility or the activity of the $P_{fliC}$ reporter in TB (Supplementary Fig. 1f), ECOR strains were not transformed with pTrc99a.

Although the majority of ECOR strains grew faster than MG1655 (Supplementary Fig. 10a) they exhibited a similar growth-phase dependence of flagellar regulon activity (Supplementary Fig. 10b). Therefore, we used the same range of $OD_{600}$ ($OD_{600} = 0.4$–$0.6$) as for MG1655 *WT* and *Ptac* to measure their expression and swimming. The activity of the $P_{fliC}$ reporter varied widely among the TB-grown isolates,

yet it was consistently below or similar to that of the MG1655 *WT* strain (Fig. 4a and Supplementary Fig. 10b, c). However, the swimming velocity of the majority of ECOR strains grown in liquid TB medium was lower than that of MG1655 *WT* or *Ptac* strains at comparable levels of $P_{fliC}$ reporter activity (Fig. 4a and Supplementary Fig. 10c). Since previous studies showed that the motility of several pathogenic *E. coli* strains[60] and other bacteria[61] can be activated when cells are grown on a surface or in a porous medium, we measured the ability of ECOR strains to spread in porous 0.27% TB agar. Indeed, the spreading of most ECOR strains, including those that were poorly motile when grown in liquid, was comparable to that of MG1655 *WT* and *Ptac* (Fig. 4b).

A possible explanation for this difference could be mechanosensitive activation of flagellar genes in cells grown in porous media or on a semi-solid agar surface, where flagella rotate under high load[60,62–64]. We therefore measured the activity of the $P_{fliC}$ reporter in cultures grown on 0.5% TB agar plates. In this case, expression in individual strains correlated well with their spreading (Supplementary Fig. 10d). While for several isolates we indeed observed an upregulation of reporter activity in such surface-grown compared to liquid-

grown cultures (e.g. ECOR-72), this was not the case for the majority of ECOR strains (Fig. 4c, Supplementary Fig. 10e and Supplementary Data 2). However, when the motility of cells grown on an agar surface was subsequently analyzed in motility buffer (see Methods for details), the population-averaged cell swimming velocity was indeed higher for many ECOR strains compared to liquid-grown cultures, now showing a dependence of population-averaged swimming velocity on expression similar to that for the MG1655 *WT* and *Ptac* strains (Fig. 4d, Supplementary Fig. 10f and Supplementary Data 2). This increase in motility was mostly due to changes in the fraction of swimmers (Supplementary Fig. 10g) whereas the swimming velocity of motile cells differed much less between liquid- and surface-grown cultures (Supplementary Fig. 10h). Thus, the observed poor motility of many ECOR isolates grown in liquid medium cannot be generally attributed to the low activity of the flagellar regulon but rather hints at some deficiency in flagellar assembly or function in liquid-grown cell. Notably, however, both flagellar gene expression and swimming of all ECOR strains were always below or comparable to that of MG1655 *WT*, suggesting that the investment in motility by natural *E. coli* isolates is under similar constraints as in the K-12 strains.

## Discussion

How microorganisms regulate the allocation of their limited cellular resources under varying environmental conditions remains an open question. Although it might be expected that gene expression levels should have been evolutionarily tuned to maximize an organism's fitness, such optimization is a multifactorial problem with mostly uncharacterized constraints and trade-offs between conflicting objectives. Particularly challenging to understand are microbial strategies for allocating resources to costly functions that do not directly benefit growth or are not used under certain conditions, which can account for up to half of cellular protein resources[13,65,66].

Here, we investigated resource allocation to flagellar motility, the most prominent of such non-growth-related cellular functions in bacteria, by titrating expression of the flagellar gene regulon and quantifying its impact on *E. coli* motility. While it was already shown that the number of flagella and cell motility increase proportionally to expression in the range below the wild-type level[27], it remained unknown whether flagella biogenesis and motility continue to increase above the native expression. By covering a wider expression range than done previously, we demonstrate that *E. coli* can increase its flagellation beyond the level observed in wild-type strains. The effect on growth fitness increases proportionally with resource investment over this wide range, too, which is consistent with the fact that flagella biosynthesis is the major component of motility costs[17,22,25]. In contrast, cell swimming velocity only increases as a function of flagellar gene expression until the number of flagella reaches ~5, which corresponds to the number observed in the wild-type MG1655 cells, but saturates above this level. This dependence of swimming velocity on the number and length of filaments was well captured by a mathematical model describing the swimming of a multi-flagellated bacterium using resistive force theory, in agreement with previous modeling of multi-flagellar propulsion[48,53,67]. Thus, the observed saturation of cell velocity appears to be the consequence of hydrodynamic constraints on *E. coli* motility.

Further supporting the general nature of this relation, not only the K-12 strains, but also the majority of motile natural isolates of *E. coli* mapped to the same unique expression-swimming relation under conditions that favored their motility. Although the activity of the flagellar regulon differed among the wild-type *E. coli* strains tested and between conditions, it was invariably confined to the sub-saturating regime. In a subset of the natural isolates, flagellar gene expression in the nutrient-rich medium was most likely selected to maximize swimming velocity as in MG1655 and other tested K-12 strains. This suggests that the regulation of motility in MG1655 is well

representative of highly motile *E. coli* strains. Such maximization of motility may be indicative of the high importance of swimming, e.g., for colonization of the environment[19,26,68]. However, even in these strains including MG1655, the expression levels remain constrained by the critical level at which swimming velocity saturates, suggesting that cells avoid unnecessary resource expenditures that provide no additional benefit. The expression levels in other *E. coli* isolates map to different points on the expression-swimming curve, covering the range below the saturation of motility. Such heterogeneity likely arises from different selection pressures on motility in the ecological niches occupied by different isolates, with motility being more or less beneficial in a particular niche. This is indeed consistent with the findings that differences in motility allow coexistence and niche segregation between *E. coli* strains, both in vitro[25,69] and in an animal host[70].

While many *E. coli* strains, including the K-12 derivatives and some natural isolates, swim similarly well when grown in either liquid or porous media, we observed that the majority of natural isolates exhibit good motility only when grown in porous or semi-solid media, possibly reflecting conditions in the animal gut. The mechanisms underlying this effect require further characterization; however, this phenomenon does not seem to be solely explained by a previously reported mechanosensitive upregulation of the entire flagellar gene regulon in porous media[60]. Many *E. coli* isolates swim poorly when grown in liquid despite having comparatively high activity of the flagellar regulon, and only achieve motility that could be expected based on their gene expression when grown on semi-solid medium. For these isolates, growth in liquid may result in the assembly of poorly functional motors or flagella. One potential mechanism for such flagellar motor remodeling in *E. coli* could be the previously described recruitment of additional force-generating units under load[62,64]. However, it remains to be seen whether this recruitment is sufficiently long-lasting to explain why these isolates retain high motility even after transfer to a liquid environment. Of note, growth in 0.27% or on 0.5% agar as used in our experiments is not sufficient to induce *E. coli* swarming, a distinct type of flagella-mediated surface motility associated with the overproduction of flagella and cell elongation that can also be observed in K-12 strains[71]. Therefore, the observed surface-dependent activation of motility does not correspond to swarming, although a potential regulatory connection between the two phenomena cannot be ruled out.

When grown under carbon limitation, *E. coli* cells exhibited similar expression-swimming and expression-cost relations in the presence of both good and poor carbon sources. However, under these conditions, the native expression of flagellar genes in *E. coli* MG1655 clearly does not maximize swimming. Instead, it seems to correlate well with the saturation of the benefit that *E. coli* could derive from chemotaxis-dependent accumulation at the source of additional nutrients, consistent with the strategy of anticipatory investment in motility under carbon limitation[25]. This contrasts with the study by Honda et al.[27], who observed only a minor difference in motility between *E. coli* cultures grown on different carbon sources, likely due to the difference between strains. Although the HE204 strain used by Honda et al. was engineered as a motile version of the originally non-motile *E. coli* NCM3722 strain[26,27] by placing the same *IS1H* insertion element as present in MG1655 upstream of the *flhDC* operon, the detailed regulation of flagellar gene expression in this engineered strain may differ from that in the naturally motile MG1655, resulting in higher basal expression of flagellar genes and lower responsiveness to the carbon catabolite repression. Such growth-related difference between these strains would not be unexpected given that NCM3722 (and its HE204 derivative) grow markedly faster than MG1655[26]. Nevertheless, our conclusions are not mutually exclusive, and it is possible that the increase in flagellar gene expression in poor carbon sources can both compensate for the decrease in the cell size and in addition result in higher motility of *E. coli* under these conditions.

The reduced activity of the flagellar regulon under carbon-limited growth revealed another prominent feature of its regulation in *E. coli*, namely the emergence of two distinct subpopulations of cells below a certain threshold of average P*fliC* reporter activity. This bimodality may be related to the recently described pulsatile activation of flagellar genes in *E. coli* at intermediate expression levels of the master regulator FlhDC[28,55]. Although this previous work concluded that the pulsatility of expression is caused by the negative regulation of FlhDC by YdiV[28], which acts as a digital expression filter[55], our experiments suggested that a downstream negative regulator FlgM, plays at least an equally important role. Furthermore, based on the established quantitative relation between gene expression and swimming motility, we could speculate on possible physiological reasons for such differentiation into distinct subpopulations. The bimodality of gene expression in microorganisms is commonly interpreted as stochastic bet-hedging behavior, which may be a better strategy in an unpredictable environment than a single adaptive phenotype[72–74]. While similar arguments were used to rationalize the differentiation of a bacterial population into motile and non-motile phenotypes[28,55,56], here we propose a different, though not mutually exclusive, explanation. We hypothesize that the bimodality, which occurs when the expression falls below the level at which two flagella per cell are synthesized and flagellar length decreases, serves to avoid the emergence of "average", poorly motile phenotypes with a single and shorter flagellum, which may be unable to benefit from motility, yet still incur the fitness cost. Such "enforced" bet hedging may provide an alternative explanation for the evolutionarily selected bimodality of gene expression, which is likely to apply not only to bacterial motility, but also to other cases where an intermediate phenotype is less fit than either of the extreme phenotypes. Thus, the hydrodynamics of flagella-mediated motility may not only determine the upper limit of swimming velocity at high levels of flagellar gene expression, but may also explain its bimodality at low levels of expression.

## Methods

### Strains and growth media

All *E. coli* strains, including natural isolates from the *E. coli* Reference Collection (ECOR)[59] and plasmids used in this study are described in Supplementary Data 1 and 2. The strain with inducer-dependent expression of *flhDC* operon (*Ptac*) was constructed previously[25] by replacing the native regulatory region of the *flhDC* operon, including the upstream *IS1H* insertion element, in the MG1655Δ*flu* background with the *tac* promoter inducible by isopropyl β-D-1-thiogalactopyranoside (IPTG). To reduce the basal expression of the *flhDC* operon, the *lacI* gene encoding the Lac repressor was additionally inserted upstream of the *tac* promoter. Deletion of the *ydiV* and *flgM* genes in MG1655Δ*flu* and its *Ptac* derivative was performed by P1 transduction from the KEIO collection[75] followed by curation of the resistance cassette by FLP recombination[76]. Deletion of the *flu* gene encoding the major *E. coli* adhesin, antigen 43, in the MG1655 group strains was used to prevent autoaggregation of motile planktonic cells[77] and thus facilitate subsequent characterization of motility[25].

To evaluate the activity of the flagellar regulon, strains were transformed with the plasmid carrying the GFP reporter for *fliC* promoter (P*fliC*) as described previously[25]. For pairwise growth competition experiments, performed as before[25], the strains were labeled by expression of either cyan or yellow fluorescent proteins (CFP or YFP) from the pTrc99a vector under the control of the IPTG-inducible synthetic P*trc* promoter[78]. Since pTrc99a carries an extra copy of *lacI*, which reduces the leaky expression from the genomic P*tac* promoter and thus the inducibility of expression in the *Ptac* strain, an empty pTrc99a vector was transformed into MG1655 *WT*, *Ptac* and other *E. coli* K-12 strains for comparability.

*E. coli* strains were grown in either lysogeny broth (LB; 10 g l$^{-1}$ of tryptone, 5 g l$^{-1}$ of yeast extract, 5 g l$^{-1}$ of NaCl), tryptone broth (TB; 10 g l$^{-1}$ of tryptone, 5 g l$^{-1}$ of NaCl), and either M9 (5× stock made with 64 g l$^{-1}$ (239 mM) of Na$_2$HPO$_4$·7H$_2$O, 15 g l$^{-1}$ (110 mM) of KH$_2$PO$_4$, 2.5 g l$^{-1}$ (43 mM) of NaCl, 5.0 g l$^{-1}$ (93 mM) of NH$_4$Cl, 2 mM MgSO$_4$, 0.1 mM CaCl$_2$, 1 µM FeSO$_4$, and 1 µM ZnCl$_2$) or Tanaka (20 mM (NH$_4$)$_2$SO$_4$, 34 mM Na$_2$HPO$_4$, 0.3 mM MgSO$_4$, 64 mM KH$_2$PO$_4$, 10 µM CaCl$_2$, 1 µM FeSO$_4$, and 1 µM ZnCl$_2$)[79] minimal media supplemented with 0.4% glucose or 15 mM succinate as the sole carbon source. Ampicillin (100 µg ml$^{-1}$) and/or kanamycin (100 µg ml$^{-1}$), and isopropyl β-d-1 thiogalactopyranoside (IPTG) were added to the media when necessary.

### Growth conditions

*E. coli* strains were grown either in batch cultures or under balanced growth. Unless specified otherwise, for the measurements in batch day cultures were prepared by 1:100 dilution of a TB-grown overnight (37 °C, 200 rpm) in 10 ml of the respective medium supplemented with kanamycin (100 µg ml$^{-1}$) for the strains transformed only with P*fliC*-GFP reporter plasmid (including all ECOR isolates). For pTrc99a-carrying strains, ampicillin (100 µg ml$^{-1}$) was also added to the medium. When minimal medium was used, cells were washed three times in medium without carbon source before inoculation. Cultures were incubated at 34 °C with shaking (270 rpm) and harvested at mid-exponential phase (OD$_{600}$ = 0.4–0.6 for TB or 0.3–0.5 for M9) measured with spectrophotometer (Ultrospec 10 cell density meter, Amersham Biosciences).

For the balanced growth, we used the same setup as described in Honda et al.[27]. Briefly, MG1655 *WT* and *Ptac* strains were streaked from stocks on LB agar plates supplemented with kanamycin (100 µg ml$^{-1}$) and, where relevant, with ampicillin (100 µg ml$^{-1}$) and incubated overnight at 37 °C. Next day, 2–3 colonies were individually inoculated in 2 ml of LB and grown for 4–5 h (seed culture). After that, 1 ml of the seed culture was pelleted (6000 rpm, 5 min), washed once in the target medium, and diluted 1:9 in the target medium. OD$_{600}$ values of the diluted samples were determined with spectrophotometer. Based on the measurements the samples were further diluted in fresh target medium (pre-culture) to the final OD$_{600}$ = 0.002 and grown overnight (34 °C, 270 rpm). The corresponding IPTG amounts were added to the pre-cultures of an inducible *Ptac* strain. In the case of TB medium, two "pre-culture" steps were performed to prevent cells from reaching stationary phase. Based on the measured OD$_{600}$ values, pre-cultures were diluted to the final OD$_{600}$ = 0.01–0.02 in 10 mL of the target medium (experimental culture). Both expression and swimming were analyzed during steady-state growth (OD$_{600}$ = 0.1–0.5).

### Reporter activity measurements

P*fliC* reporter activity was measured by either flow cytometry or plate reader assay. For flow cytometry, cultures grown in batch or under balanced growth conditions were diluted 1:50 in tethering buffer (6.15 mM K$_2$HPO$_4$, 3.85 mM KH$_2$PO$_4$, 0.1 mM EDTA, 1 µM methionine, 10 mM sodium lactate, pH 7.0) and fluorescence was detected using a 488 nm laser (100 mW) and a 510/20 nm bandpass filter for GFP on a BD LSRFortessa SORP cell analyzer (BD Biosciences, Germany). 30,000 individual events were analyzed in each experimental run. Gating was first performed on an FSC-A/SSC-A plot and on an SSC-W over SSC-H plot to exclude doublets[80]. Events in the samples with fluorescence intensities higher than the background signal from the MG1655 *WT* or *Ptac* strain without the reporter plasmid were considered 'positive' (Supplementary Fig. 8). The proportion of 'positive' events per sample and summary statistics (mean, median fluorescence values) of both the 'positive' and the 'whole' population were assessed during the measurements using BD FACSDiva™ Software v8.0.1. Data were collected in FCS 3.0 file format and analyzed using the flowCore package in R v4.2.2.

For growth and expression measurements in the BioTek Synergy H1 plate reader. Unless specified otherwise, overnight cultures were inoculated into the 96-well plates (Greiner Bio-One) at a dilution of

1:1000 and grown at 34 °C with double orbital shaking at a frequency of 548 cycles per minute (CPM) and a shaking amplitude of 2 mm for 24 h (TB) or for 48-64 h (M9). GFP fluorescence was quantified using a monochromator-based filter set (excitation 485 nm, emission 530 nm, with a bandpass ≤18 nm for detection). Fluorescence and optical density ($OD_{600}$) were measured every 10 min. Since the experiment with ECOR strains was performed independently using a different fluorescence detector gain, the GFP/$OD_{600}$ signal of MG1655 *WT* reaches a higher value on Supplementary Fig.10 compared to Supplementary Fig.1. For experiments shown in Supplementary Fig. 7, the TECAN Infinite M1000 PRO plate reader was used instead for consistency with the previous study[25].

Reporter activity in ECOR isolates was also measured after growth in liquid TB medium in flasks or on the surface of semi-solid TB agar (0.5%). For the liquid medium setup, day cultures were prepared in the same manner as batch cultures of MG1655 *WT* and *Ptac*. For growth on the semi-solid medium, 20 μl of the same overnight culture was spread on the surface of TB agar using glass beads. After drying for 15–20 min, the plates were incubated at 34 °C for the same time as it took the strain to reach $OD_{600}$ = 0.4–0.6 (i.e., 2.5–4 h) in liquid medium. Cells were gently washed from the plates with 2 ml of motility buffer (6.15 mM $K_2HPO_4$, 3.85 mM $KH_2PO_4$, 0.1 mM EDTA, 67 mM NaCl, pH 7.0) and adjusted if necessary to final $OD_{600}$ = 0.5, and 1 ml of a liquid-grown culture was also washed once in motility buffer. After another washing step, the cells were resuspended in 1 ml motility buffer supplemented with 1% glucose and 0.001% Tween-80. GFP fluorescence was measured in a TECAN Infinite 200 PRO plate reader at 480 nm wavelength, 9 nm bandwidth for excitation and 510 nm wavelength, 20 nm bandwidth for emission.

### Analysis of swimming velocity and flagellar rotation

Bacterial cell motility was analyzed as previously described in refs. 25,81. Briefly, 1 ml of the same cell culture as prepared for flow cytometry was gently centrifuged (1500 g, 5 min), washed twice in motility buffer, and resuspended in 1 ml motility buffer supplemented with 1% glucose and 0.001% Tween-80. 3–5 μl of this cell suspension was introduced into a custom-made chamber between two coverslips, forming a cylindrical drop of ~500 μm height and ~3 mm diameter. Motility was imaged at least 100 μm away from all the edges of the drop to avoid boundary effects, by phase-contrast video-microscopy (Nikon TI Eclipse, 10x objective with NA = 0.3, Phase 1 ring, CMOS camera EoSens 4CXP), with 10,000 frames being recorded at a rate of 100 frames per second (fps). Motility parameters, in particular the fraction of swimming cells and the swimming velocity of swimmers, were extracted from the movies using differential dynamic microscopy (DDM)[29] (see Supplementary Note 1).

To determine the frequency of flagellar rotation, samples were prepared in the same manner as described for swimming velocity analysis. A 10,000-frame movie with a field of view of 512 × 512 $px^2$ (1 px = 0.7 μm) was acquired far from the sample surfaces under dark field illumination (Nikon TI Eclipse, 10x objective with NA = 0.3, CMOS camera EoSens 4CXP) at a rate of 800 fps. Dark field illumination is obtained by combining an aligned Ph3 condenser ring with the 10x objective on the Nikon TI Eclipse microscope. All data were analyzed using the dark field flicker microscopy (DFFM) method[50] (see Supplementary Note 1) implemented in ImageJ (https://imagej.nih.gov/ij/) with custom-written plugins. The plugins for DDM (previously published) and DFFM are available at https://gitlab.gwdg.de/remy.colin/FourierImageAnalysis. Briefly, DFFM uses the flickering that results from changes in the direction in which light is scattered by anisotropic objects as they rotate to measure the rotation speeds of the cell body and flagellar.

### Motility assay in soft agar

Motility driven spreading of *E. coli* in 0.27% TB soft agar was analyzed as previously described in ref. 60. Briefly, 2 μl of overnight cultures grown in TB (37 °C, 200 rpm) were transferred to the soft agar plates, and the diameters of the spreading zones were measured after 4–5 h of incubation at 34 °C by capturing images with an iPad camera and quantifying the diameter of the spreading zone using ImageJ.

### Pairwise growth competition

Growth competition assays were performed as previously described in ref. 25. Briefly, the overnight cultures of the MG1655 *WT* or *Ptac* strain expressing CFP and the *ΔflhC* strain expressing YFP, grown individually in TB (37 °C, 200 rpm), were mixed in a 1:1 ratio to final $OD_{600}$ = 0.0025 in 2.5 ml of fresh media and cultured for 24 h (TB) or 48-72 h (M9 minimal medium) at 34 °C and 200 rpm. The expression of YFP and CFP was induced with 10 μM IPTG for the co-culture containing the MG1655 *WT* strain or by the corresponding IPTG concentrations used for induction of the chromosomal *Ptac* promoter. For the chemotactic benefit assay, differentially labeled non-chemotactic *ΔcheY* strain and MG1655 *WT* or *Ptac* strains were grown in Tanaka minimal medium for 72 h without shaking in the presence of nutrient gradients generated by 40 μl volume agarose beads (2% agarose) containing 12% casein hydrolysate as described previously[25]. The initial and final proportions of CFP- and YFP-labeled cells were measured by flow cytometry on the BD LSRFortessa SORP cell analyzer (BD Biosciences). The sample was excited with lasers at 447 nm (75 mW), 514 nm (100 mW), and 488 nm (20 mW), with the latter used to identify all cells. CFP and YFP emission signals were detected at 470/15 nm and 542/27 nm, respectively. The fraction of CFP/YFP-'positive' events per sample was assessed during the measurements using BD FACSDiva™ Software v8.0.1. Summary statistics were collected in csv file format and analyzed in R v4.2.2.

### Measurements of flagellar length and number

For flagella staining, 1 ml of the mid-exponential cell culture grown in TB as described above was centrifuged (3000 g, 3 min) and gently washed three times in Buffer A (10 mM $KPO_4$ buffer, 0.1 mM EDTA dipotassium salt, 67 mM NaCl, 0.001% Tween-80, pH 7.0). The cell pellet was resuspended in 400 μl of Buffer B (same as Buffer A but adjusted to pH 7.8 with $NaHCO_3$), and 8 μl of 10 μg ml⁻¹ Alexa Fluor 594 succinimidyl ester dye dissolved in DMSO was added to the mixture. Samples were incubated at 30 °C in the dark with gentle shaking (100 rpm) for 90 min, washed three times in Buffer A and diluted fivefold in Buffer A. 3–5 μl of cell suspension was applied to a 1% agarose pad (in tethering buffer) and transferred to a 2-well μ-Slide (ibidi, Germany).

Fluorescence widefield images were acquired using a Zeiss Elyra 7 inverted microscope with a 63x oil/1.46 oil objective and a further 1.6X magnification. The sample was excited with a 561 nm 500 mW laser (1% power) using a quadruple band dichroic and emission filter. The fluorescence emission of the succinimidyl ester was detected at 595/50 nm interval with a PCO 4.2 Edge sCMOS camera, the exposure time was 100 ms. The number of flagella was quantified for 106 cells in total from multiple fields of view per condition, including both flagellated and non-flagellated cells. The length of flagellar filaments was measured using segmented line tool of ImageJ.

### Immunoblot analysis of intra- and extracellular flagellin

To shear flagellar filaments, a 1 ml aliquot of the mid-exponential cell culture was passed through a 1 ml syringe with the 26 G needle 20 times, and centrifuged at 2500 g for 10 min. The supernatant and cell pellet, resuspended in 333 μl of TB medium, were further analyzed by immunoblot. To transfer the samples to the membrane after SDS-PAGE, a PerfectBlue Semi-Dry Electroblotter (Peqlab, VWR, Germany) was used at constant amperage for 1 h (150 mA for 8*6 cm membrane and 1.5 mm thick gel). After transfer, the membrane was stained with Revert™ 700 Total Protein Stain for Western Blot Normalization (LI-COR Biosciences, Germany) and, after blocking, incubated overnight

(4 °C, orbital shaking) with the anti-flagellin rabbit polyclonal FITC-conjugate primary antibody (AA 2-498-FITC, ABIN2831532, 1.5 mg/ml) diluted 1:10,000. Next day, IRDye 800CW donkey anti-rabbit IgG secondary antibody (LI-COR Biosciences; P/N: 926-32213, 1 mg/ml) was added to the membrane at a dilution of 1:10,000. Fluorescence was measured using an Odyssey Clx Infrared Imaging System (LI-COR Biosciences, Germany) in two channels (700 and 800 nm). Images were analyzed and processed using ImageJ.

**The model of flagellum-mediated bacterial swimming**

The model for multiflagellated propulsion extends the classical force balance analysis for uniflagellated propulsion[41,82] and accounts for our measurements of swimming speed, cell body rotation speed, and flagellar rotation speed, as well as flagellar length, flagellar number, and cell size. The model is described in detail in Supplementary Note 2. Briefly, we assume that the multiple flagella form a single tight bundle[83,84], described in the framework of resistive force theory[43,44,82,85] as a rigid helix of larger thickness for a higher number of flagella, supported by macroscopic experiments at low Reynolds number with multiple helices[86,87]. We account for the increase in both flagellar length and flagellar number with increasing *flhDC* induction. The cell body is described as a counter-rotating rod[45,46] of fixed size, consistently with our observation. The flagellar motor speed is assumed to be constant, in agreement with our measurements of the flagella and cell body rotation speeds. The balance of forces and torques acting on the cell body and the flagellar bundle provides predictions of the swimming speed and the rotation frequencies. All model calculations were performed using Matlab R2020a. The code used is available at https://gitlab.gwdg.de/remy.colin/RFTMultiflagella.

**Reporting summary**

Further information on research design is available in the Nature Portfolio Reporting Summary linked to this article.

## Data availability

All data are available in the main text or in Supplementary Information. Source data are provided with this paper.

## Code availability

Matlab codes for the model are available at https://gitlab.gwdg.de/remy.colin/RFTMultiflagella. The plugins for DDM (previously published) and DFFM are available at https://gitlab.gwdg.de/remy.colin/FourierImageAnalysis.

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

## Acknowledgements

We thank Julian Pietsch and Santiago Kuhl for fruitful discussions and feedback on the manuscript. We thank Silvia Gonzalez Sierra and Gabriele Malengo for the technical assistance with flow cytometry and microscopy experiments, and Irina Kalita for the help with flagella labelling. This research was funded by the Max-Planck-Gesellschaft and by the Max Planck School Matter to Life supported by the German Federal Ministry of Education and Research (BMBF).

## Author contributions

I.L., B.N., and V.S. designed the study. I.L., R.C., B.N., and V.S. designed the experiments. I.L., R.C., H.Y., and B.N. performed the experiments. I.L., R.C, and H.Y. analyzed the data. I.L., R.C., and V.S. wrote the manuscript.

## Funding

## Competing interests

The authors declare no competing interests.
