## [Transparent Peer Review file · Nature Communications]

Physics of swimming and its fitness cost determine strategies of bacterial investment in flagellar motility

Corresponding Author: Professor Victor Sourjik

Version 0:

Reviewer comments:

Reviewer #1

(Remarks to the Author)

In this manuscript, the authors titrated the expression of the flagellar gene regulon (placed under the control of an inducible promoter) and demonstrated that the activity of the flhDC (master regulator) operon has a limited physiologically relevant range: the average swimming velocity of the cells plateaus when the flhDC operon activity is induced to wild-type levels and further increasing the operon activity does not increase swimming velocity. Through a physical model, the authors argue that after reaching five flagella — the average number found in wild-type cells — adding more flagella does not increase swimming velocity. The number of flagella depends on flhDC operon activity, indicating that the hydrodynamics of flagellar propulsion limits the physiologically relevant range of flhDC operon activity. Furthermore, the authors tested several wild *E. coli* isolates and found that when grown on agar, their swimming velocity also plateaus when their flhDC operon activity reaches an activity level close to that of WT MG1655.

This is a compelling result, and the experiments in this paper are very carefully reported. The main suggestion I have is to more clearly delineate the main conclusions of this work from a recent related work by Honda et al. (PNAS, 2022). In the latter study, Honda et al. reported a correlation between swimming speed and PflIA activity, which also saturates at wild-type induction levels (Figure 2 of their paper). Since FliA is a transcription factor that directly controls fliC expression, PflIA activity is closely related to PFLiC activity. Moreover, Honda et al. also demonstrated that the number of flagella depends on PflIA activity, and that motility is maximized when *E. coli* possesses approximately five flagella (Figure 4, panels E and F of that paper).

Some specific concerns regarding the experiments and various aspects of data presentation (which I believe can be significantly improved) are listed below.

Major:

Reporter Activity Measurements: The authors state that they harvest cells at an OD between 0.4 and 0.6 for TB or 0.3 and 0.5 for M9. However, it is known that tumble bias and swimming velocity of cells (Staropoli and Alon, Biophys. J, 2000) depend on the harvesting OD. Could the rather broad range of harvesting ODs in the present study explain the substantial variability (at least as large as the range of swimming velocity changes predicted by the physical theory) observed between WT replicates reported in Figure 1d?

Figure 2d: The swimming velocity predicted by the RFT model only varies between approximately 20 and 24 $\mu\text{m}/\text{sec}$, suggesting the minimum achievable speed with one flagellum is 20 $\mu\text{m}/\text{sec}$. However, the minimum measured swimming velocity is reported to be 5 $\mu\text{m}/\text{sec}$ in Figure 1d. What accounts for this discrepancy? Could the expression of other flhD-dependent genes be relevant? The validity of this physical model would be more convincingly demonstrated if the authors could experimentally verify the conclusions of their RFT model with experiments at the single-cell level. It would seem to be technically feasible to assess the swimming velocity of cells whose flagellar filament number are counted by staining with a fluorescent dye.

Figure 3a: The activity of the fliC promoter in WT cells is higher when succinate is used as a carbon source compared to glucose due to catabolite repression. However, this pattern is also observed for the expression of the flhDC operon in the inducer-dependent strain. Since the native regulatory region of the flhDC operon is absent in this strain, why does it still

undergo catabolite repression?

Minor:

Figure 1b: The x-axis labels are confusing; I recommend using IPTG concentrations instead, similar to what is done for Figure 4b. The WT condition can be indicated with a broken x-axis. Additionally, the y-axis label is simply "fluorescence," whereas in panel c, the mean value of the same quantity is labeled "PfliC activity." I suggest the authors reconsider these labels for consistency. The same applies to Figure 3.

Figure 1d: In the figure legend, MG1655 is denoted as "WT", while RP437 and W3110 are not; aren't all these strains wild type for chemotaxis?

Figure 1e: The y-axis label is confusing, "% of WT/Ptac" can be interpreted as a ratio of WT and Ptac cells, but what the authors are reporting is the fraction of WT or Ptac cells compared to Δ FliC cells.

Figure 3c: The y-axis label is confusing. What the authors call "positive population" essentially refers to the number of cells with PflilC reporter fluorescence intensity above an arbitrary threshold (as shown in Extended Data Figures 8 and 9). A 100% positive population corresponds to a fully non-bimodal population. I recommend changing the axis label to something like "degree of bimodality" and adding an inset with a cartoon (or actual data) in the white space of this figure to explain how this quantity is defined.

Figure 3c: In the figure legend, the IPTG concentration for the Ptac induction is not specified, is it correct to assume that this is at zero induction? This is what line 180 of the text suggests.

Figure 4: In multiple axes labels the authors use the notation "(L)" and "(S)" after "PflilC activity" to denote whether promoter activity was measured in liquid or agar. In the rest of the paper, the authors use a similar notation to report whether they used a flow cytometry or plate reader assay. I recommend using consistent notation throughout, for example, "PflilC activity (PR) in Liquid."

Figure 4b: The x-axis labels after "MG1655 WT" are shifted.

Extended Data Figure 4: To enhance data presentation, the authors should extract the fluorescence intensities from their western blot and include a plot of the FliC and total protein amount as a function of the induction level. For instance, 3 μ M IPTG seems to induce WT levels of FliC, while in Figures 1b and 1c, it is reported that 10 μ M IPTG induces WT-like levels of PflilC activity.

Reviewer #2

(Remarks to the Author)

Reviewer #3

(Remarks to the Author)

The manuscript "Physics and physiology determine strategies of bacterial investment in flagellar motility" by Lisevich et al. reported a study that investigates the relation between the expression of motility genes and the swimming speed and the growth fitness of *E. coli*. The main finding of the manuscript is that hydrodynamics is the limiting factor regulating the motility gene expression. While it is interesting that the authors created a strain of *E. coli* with a titratable expression of the flagellar regulon, the conclusion of the manuscript is neither sufficiently novel nor well supported by the theory. In addition, we found the writing of the manuscript is confusing including experiments and discussions that may not be directly relevant to the main conclusion and theme of the work. Thus, unfortunately, we cannot recommend the publication of the manuscript in *Nat. Commun.*

We listed our comments and concerns below, which we hope can be useful for the authors to revise the manuscript.

1) For the fitness measurements (Line 97-104): Although Ptac strains are derived from MG1655 WT, we are wondering if the growth rate of the strains is affected by genetic engineering. A control experiment should be conducted, which compares the growth rate of the Ptac strains with that of MG1655 WT independently without co-culturing. In addition, does the presence of two strains affect their natural growth rates?

2) As the major conclusion of the manuscript, the authors argued that the increase in the number of flagella does not increase the swimming speed. While this work provides fitness measurements in co-culture as a new way to assess the effect, such a conclusion has already been well established (see e.g. the early discussion by Darnton et al, *J. Bacteriol.* 189, 1756–1764 (2007) and the theoretical/numerical work by Nguyen & Graham, *Phys. Rev. E* 98, 042419 (2018)), which

certainly diminishes the novelty of the work.

3) Even within the current work, there are a couple of issues regarding the hydrodynamic theory developed by the authors. First, it is well known that the flagellar motor operates with constant torque, instead of constant rotation speed, during the normal run phase of bacteria (e.g. Berg, *E. coli in Motion*). However, the experiments by the authors contradict the established results. How do the authors reconcile the difference? Second, Fig. 1d shows that the swimming speed decreases at the high expression level and exhibits a non-monotonic trend. Can the model also explain the decrease of the swimming speed with the increase of the flagellar number at large flagellar numbers?

3) Indeed, it is an open question why bacteria grow multiple flagella, as the number of flagella does not increase the swimming speed of bacteria. Many other possible functional benefits have been discussed (e.g. the increase of swimming stability or the enhancement of surface attachment). We would suggest the authors to survey the extensive literature on the topic more thoroughly. It is hardly convincing based on a single observation and a simplified (maybe questionable) calculation to conclude that the hydrodynamics is the sole factor limiting the growth of more flagella.

4) The authors showed that the motility gene expression affects both the number and the length of flagella. We are wondering if other properties of flagella (e.g. pitch angle, helical radius et al) are also affected and how they impact the observed behavior.

5) The authors argued that "native gene expression under carbon-limited growth might correlate with the potential benefit that could be achieved in a given carbon source by performing chemotaxis towards sources of additional nutrients". It is far from clear how the expression of motility genes (or the number of flagella) affects chemotaxis. It seems that the authors tried to distinguish swimming speed and chemotaxis as two independent effects. Maximizing one does not affect the other. In fact, the swimming speed and chemotaxis are closely related. A lower swimming speed leads to poor chemotaxis. After all, swimming evolved to achieve chemotaxis.

6) It is not clear how the two subpopulations are related to the main conclusion of the manuscript, i.e., hydrodynamics determines the motility gene expression. Some clarification of the relation between the two discussions is necessary, otherwise the discussion is a distraction of the main theme of the paper. Moreover, the authors argued that "the negative regulation of FlhDC activity by YdiV" is not the origin of bimodality. Then what is the origin?

7) Swimming in bulk fluid and swarming on agar surface or in porous media are quite different from the hydrodynamic standpoint of view (see e.g. Kearns, *Nat. Rev. Microbiology* 8, 634–644 (2010)). So the discussion relating the locomotion of bacteria in the bulk fluid and along the surface of agar is quite confusing. It is not clear if the conclusion reached by the swimming in the bulk fluid via hydrodynamics can be translated to the swarming in agar plate. More evidence is needed to support the hypothesis on "some deficiency in flagellar assembly or function in liquid-grown cell".

8) While it is reasonable that different niches and environments lead to different saturated numbers of flagella, why does the MG1655 WT provide the upper limit of all different strains?

Minor points:

1) In Line 43, the authors stated, "Motility consumes several percent of total cellular resources ..." Could the authors be more specific and clarify the range of the value?

2) In Line 91, the authors claimed that the two other derivatives of *E. coli* showed a similar relation. Nevertheless, from Fig. 1d, the two derivatives have fixed gene expression without showing the non-monotonic trend. So it is unclear what the similar relation means here.

Reviewer #4

(Remarks to the Author)

Reviewer #5

(Remarks to the Author)

The authors study motility regulation in *E. coli*, but because they do not put their work well in the context of the existing literature the manuscript is difficult to follow. As far as we understand they build on the work from Honda et al PNAS (<https://www.pnas.org/doi/10.1073/pnas.2110342119>) that shows exponentially grown *E. coli* upregulated gene expression in nutrient-poor environments to maintain the same flagella number per cell because the cell size decreases. The swimming speed and motile fraction thus stay roughly constant across growth conditions (i.e. both nutrient-poor and rich growth media). In the Honda et al PNAS study authors titrate the expression of flagellar genes and show that the maximum expression they can achieve corresponds to the level observed in WT and to 4-5 flagella per cell, where roughly that number is needed to achieve wild-type, exponential growth motile fraction and swimming speed. Here authors achieve expression above the WT level and show that at that level of expression, although the cells make more filaments, they do not swim faster, limited by the hydrodynamics of swimming.

For this result to be of interest to the general audience of Nature Communication the authors would need to demonstrate that what they observe is a general observation.

To start, why is there a difference between the authors' observations and those of Honda et al mentioned above?

One explanation would be that Honda et al simply did not induce above the WT level so have never seen that expression goes further up, this seems unlikely though (is it?).

Other differences between this manuscript and previous work are the strains (here MG1655, and in Honda et al NCM strains) and the stage of growth (exponential phase in Honda et al, and undefined state here).

The authors would need to perform experiments that tell us what the source of the difference is. To check if it is the strain authors would need to compare NCM and MG1655 strains in exponential phase. If it is a strain difference, and it is known that NCM has an issue with RpoS authors should then check if NCM strain repaired for RpoS behaves as they see for MG1655, all in exponential phase. If this is the case, their results are of interest to a general audience of Nature Communications, because this would indicate that what they observe here is more what we should expect from wild-type strain than what Honda et al observed, with NCM strains defective in RpoS.

If it is not, if it is simply the stage of growth, then this result is more suited for a specialised audience journal, as it depicts what they see when they grow the cells in the specific way that they do. And that state is not well defined.

To expand on the stage of growth. Here the authors are growing the cells in Tanaka media with defined carbon sources, as well as in TB and LB where cells are consuming one preferred amino acid after the other in series, and as soon as they run out of the most preferred one. They also dilute their culture o/n, either 1x100 or 1x1000, which is not enough for cells to get to the exponential phase (they would have not fully diluted from the o/n stationary phase in the little generations they can achieve without running out of some nutrient). In Honda et al, cells are heavily diluted at low ODs to ensure they do not run out of any nutrient and reach at least 6 (but most of the time more) generations in such a constant environment.

Not to say that studying other stages of growth is not interesting, in fact, bacteria are hardly ever in such a well-defined exponential phase. The problem is that the state authors have them in is not well defined. The exponential phase means cell number, biomass, and even individual cells are all growing at that exponential rate, with the previous phase well diluted. We know from 'Copenhagen school', i.e. the older papers from Schaechter, Maaløe, Kjeldgaard, etc that the way cells exit the exp phase (i.e. what changes in physiology first) depends on what has run out in the media. Leaving us with interesting, but not well-defined growth phases. And, what happens there is at this stage of more interest to the specialist audience (until we find a way of better defining those growth phases).

The authors complement their experimental findings with theoretical predictions based on the Resistive Force Theory, which we found well-written (and service to the community to have the derivation as well explained as here), but also found some mistakes. We believe these are important because it will ensure authors put less emphasis on 'excellent agreement' between RFT and their data. Once we corrected the found problems we did not see such a good agreement. For their arguments, this is not important, but it is for estimates on how good of an approximation RFT is (not as good as authors seem to argue).

Following are some comments on specific lines in the manuscript (including minor typos etc) for authors to consider:

Title -suggest to change. Currently, it does not mean much

The abstract is not very clear and we suggest authors make it punchier and easier to understand.

Line 14, costly needs to be put in context. It's cheap to power flagellar motility from the point of view of cellular energetics, but expensive to make it from the point of view of protein allocation

Line 15-17 it's a bit of an awkward phrasing, suggest revising the sentence

Line 18-19 it's not obvious what the authors mean by 'physiologically relevant range of investment' before reading the manuscript. It would be useful to revise. The rest is unclear as well

Line 12 a typo 'for different functions' rather than in (although happy for authors to leave this to be corrected by the journal staff and given the publication charges). We would not agree with 'poorly' either as there has been a decent amount of work on this recently, we'd simply say that it is not fully understood yet.

Line 30 typo, capital T at the beginning of the sentence (as above when it comes to typos)

Line 44, some of the key references for this are missing, such as <https://pubmed.ncbi.nlm.nih.gov/26632588/> (protein reallocation)

<https://pubmed.ncbi.nlm.nih.gov/21390082/>

<https://pubmed.ncbi.nlm.nih.gov/17438023/> (growth curves with and without FlhDC)

Line 45, there is a key figure for this claim in ref <https://pubmed.ncbi.nlm.nih.gov/26632588/> as well

Line 45-48 This is confusing and is citing papers that claim different things as if they claim the same (to our understanding of these papers). For example, the following paper <https://www.pnas.org/doi/full/10.1073/pnas.1910849117> by some of the authors, claims that increased gene expression in poor conditions gives an advantage because the cells can follow nutrient gradients in those conditions (i.e. have a chemotactic advantage because the swimming speed and chemotactic drift velocity increase). Whereas this publication <https://www.pnas.org/doi/10.1073/pnas.2110342119> says that an increase in gene expression in poor nutrient conditions is there simply to have a constant flagella number as cells become smaller. And, that cells' swimming speed and ability to do chemotaxis is not dependent on the growth rate so that cells have that swim and chemotax in nutrient-rich conditions expand faster than those in poor <https://pubmed.ncbi.nlm.nih.gov/31695195/>. So to group it all under one as 'trade-offs' is misleading.

Line 49-56 This now explains better my comment above. But I am leaving it, as that information was needed before. I think the authors should do a better job of explaining this straight away (in fact maybe those last sentences, lines 45-48 are not needed, and instead this paragraph should simply follow, where the difference in explanations is made clearer).

Line 57-58 But this has been done in all those publications we are discussing above, and it leads to opposing opinions. So how is what is being done now different? And what is the main reason these experiments are now done? What exactly do the authors wish to investigate given what is given in previous literature? All this should be made clear.

Line 59-64 It is not really clear from this what the authors learn new, as opposed to what is in agreement with previous literature.

Line 69 and the paragraph that follows. In light of the previous literature (for example, figures 1 of both of the above PNAS

papers) it is not clear why the authors are doing this? And how the results are similar and different from what has been done before. Please put the results in a better context of previous literature so that the novelty of the current results is clear. The authors also mention different strains and include them. But, as mentioned above the most relevant previous publication for this work uses NCM strains (and those can have an issue with RpoS), and that has not been included. It is great that the authors are using RP437 (and W3110), because that strain is frequently used for swimming as they say. But there are hardly any other physiological studies on it, without doing all of those, this comparison here is not as informative.

Line 75: double "in" typo.

Line 106 and corresponding results chapter. While the authors describe their results well, these have not been put in the context of existing literature.

For example, <https://www.pnas.org/doi/10.1073/pnas.2110342119> Figure 4E or S

Line 122-123 We disagree with this statement. The flagella are not at the pole like in RFT, but randomly distributed along the cell body. Thus, the torque the motor is producing is perpendicular to the length of the swimming. It is not at all clear therefore that this approximation is the best for understanding the hydrodynamics of E.coli swimming. It seems to be simply what we can do now.

Line 124-125 please add a bit more details on the assumption (e.g., and importantly, all the motors are at the back of the cell body)

Line 125-130 We think that the fact the authors see that the motor speed is independent of the flagella number is a result in itself and the authors should put more emphasis on it. But, we disagree with the interpretation that follows the description of this result. The zero-torque speed of the E. coli motor is reported close to or higher than 300 Hz

(<https://www.sciencedirect.com/science/article/pii/S0006349500766628>, or

<https://www.ncbi.nlm.nih.gov/pmc/articles/PMC5703321/> or) at room temperature in the presence of a good carbon source.

220 Hz is not far from the 'knee' (<https://www.sciencedirect.com/science/article/pii/S0006349500766628>) of the torque

speed curve and not much faster than the rotation speed of a 0.5µm bead in those conditions, i.e. not zero-load

([https://www.cell.com/biophysj/pdf/S0006-3495\(19\)30392-3.pdf](https://www.cell.com/biophysj/pdf/S0006-3495(19)30392-3.pdf)). Thus, the motor speed should change if the load changes.

The fact that it doesn't with an increasing number of flagella suggests that the torque supplied by each motor is roughly constant, and thus that the increase in load due to larger flagellar length and bundle thickness is balanced by the fact that more motors share that load, hence placing each motor under a roughly constant load as a function of N. We noticed that the authors did mention this as a side note, in Line 137-139 of the SI. We think it's a major contributor.

Linked to this, we suggest removing Figure 5g unless it can be proven experimentally because it is formulated for a single perfect helix (i.e. perfect, tight bundle) and neglects all possible interactions between multiple filaments. Furthermore, by rearranging $\omega_m = \omega_f(N) - \omega_b(N) = \text{cst}$, and $\Gamma_f = \mu_{RR} \omega_f(N) - \mu_{TR} U(N) = -\Gamma_b = -\mu_{RR} \omega_b(N)$, one obtains the decay of bundle rotation speeds seen in ED Fig.5c (although with slightly higher frequencies see note below), thus the assumption that the motor torque decreases roughly as $1/N$ isn't needed to explain the experimental results.

Line 130-132 Why not include the experimental data in Fig. 2d? Please also cite previous literature and acknowledge that it shows the same with slightly different parameters

<https://www.sciencedirect.com/science/article/pii/S0006349595800895?via=ihub>, (Fig.7b)

Line 131-138 While the model predictions are in agreement with experiments authors put a too strong emphasis on this given that they have made a mistake in the ratio of the drag coefficients (in SI calling it 'the agreement is overall excellent'). As a result, they think RFT is overall a better predictor of E.coli swimming speeds than it might be (we think it's simply indicative). The γ_k parameter in Eq 2.16 of the supplementary note is the ratio of the tangential to normal drag coefficients in Lighthill's analysis. It is not constant when the thickness of the effective filament varies. Furthermore, the value of 0.7 was obtained for sea urchin spermatozoa, not bacteria. Using the coefficients derived by Lighthill in Flagellar Hydrodynamics (1976), $K_n = 4\pi\eta / (\ln(0.18\Lambda/r_f) + 0.5)$ and $K_t = 2\pi\eta / \ln(0.18\Lambda/r_f)$ with $\Lambda = \lambda / \cos\psi$, we obtain significantly higher swimming speeds than the ones predicted in the manuscript. Using a coefficient of 0.25 (as in the manuscript, not clear why?) instead of 0.18 (as above) doesn't affect this. It also seems that the authors consider a diameter of 40 nm for an individual flagellar filament, but we think values around 20 nm are more common, suggest to double check. Authors should update the predictions with the correct parameters. We note that RFT obtained speeds larger than those observed in experiments would not be surprising, since RFT doesn't account for hydrodynamic interactions. It would then also be good to get the flagella rotation speed from the model for a given motor speed in ED Fig. 5c?

Line 143-144 and the corresponding results chapter. Here again, we feel the results are well described and need to be put in better context with previous literature. The result of Honda et al in PNAS paper Figure 2C is similar to figure 3b here, however, there cells grown in different carbon sources reach the same speed that corresponds to the maximum gene expression level. Here, they don't, and the average swimming speeds are lower for WT in media with lower nutritious value. This is consistent with the following results by some of the authors

<https://www.pnas.org/doi/10.1073/pnas.1910849117> (specifically one of the SI figures there) But here again, the main

difference seems to be the way the cells are grown. Following previously published figure

<https://www.nature.com/articles/s41586-019-1733-y/figures/6> makes this explicit. So we are not sure what these results add?

If the cells are grown in the particular way the authors are growing them, and that is not a well defined state, the result is the same across many different media? This could be interesting as it indicates that the way authors are growing the cells is a 'state' in the physics meaning of the word. But if that is the point this section needs rewriting, and also plotting in a slightly different manner (maybe all the swimming speeds obtained in different media when cells are grown like the authors grow them need to be plotted against the growth rate at the time the cells were collected).

Line 154-159 Yes, but it disagrees with the other two studies as mentioned above and the authors are not adding anything to explain the disagreement.

Line 166-168 Or it is simply a consequence of not properly diluting yet from o/n growth. Needs checking whether the bimodality would still be there if cells were grown in exp phase. Otherwise, it's just an observation for the cells that are grown in a particular way of growing cells.

Line 174-177 but neither seems enough. Can the authors calculate how many generations the cells achieve with 1×100 and

1x1000 dilution at their growth rate to the OD they harvest them at? We suspect $10^4/10^5$ dilution will be needed to get 6-10 generations.

Line 180: ref to Fig.1b, not 1c.

Line 196 and corresponding results section. This is potentially very interesting, but it's hard to interpret because cells were again not grown to the exponential phase (and the way they were grown could be a different state for each different isolate). It either needs growing into exponential phase or a better understanding of the current state the cells are grown in. Motile fractions and the swimming speeds of motile cells should also be provided for those experiments. At the moment it is unclear if it is the motile fraction or the speed of the motile cells or both that change when cells are grown first on semi-solid medium. Line 335-337: everything should be written in mM for the minimal media to allow comparisons between the two. Also, in the original paper Tanaka medium contains 20 mM $(\text{NH}_4)_2\text{SO}_4$, 34 mM of NaH_2PO_4 and 64 mM of K_2HPO_4 . Typo or modified medium?

Line 386 that is a very small volume for DDM experiments. What is the thickness of the sample, and how far away from the glass slide are the cells imaged? Can the authors exclude wall effects?

Supplementary Note Line 93-94. Yes for the thrust, but in ref 10 they show a major drop in hydrodynamic efficiency if helices are out of phase, because the drag on resulting propeller is increased and thus the speed decreases. So, the bundle configuration does matter.

Supplementary note, RFT section: typo when $\|$ is used instead of index b for body coefficients?

Reviewer #6

(Remarks to the Author)

Version 1:

Reviewer comments:

Reviewer #1

(Remarks to the Author)

The authors have addressed all of my requests from the previous round, as well as many comments/suggestions from the other reviewers. As a result, the manuscript is now substantially improved.

Most notably, they now much more clearly explain the difference between the present study and that of Honda et al. (2022, PNAS). Whereas the Honda paper only titrated expression of the flagellar regulon below WT levels, the present study used an expression system capable of increasing the expression of motility genes above WT levels, to clearly confirm that motility levels saturate very close to the wildtype expression level of the regulon. Additionally, they now clarify that the Honda et al. study used a strain (HE204, derived from NCM3722) that is not naturally motile and hence has not been evolutionarily optimized for motility. This distinction is significant in considering the generality of the results, which are further strengthened by additional experiments with wild isolate strains. The revised text emphasizes these differences well enough.

The authors also made substantial improvements in data presentation, addressing all my previous concerns. The data is now much clearer, and improved data presentation for validation of their RFT model (Figure 2e).

The authors also performed additional experiments to address the origins of bimodality in the expression of flagellar genes. They now argue that both YdiV and FlgM contribute to that bimodality.

Finally, they also tested the effect of balanced exponential growth on flagellar gene expression. Instead of using cells recovering from saturated stationary-phase cultures, they seeded their experimental cultures with "pre-cultures" that had not reached saturation. This approach mirrors the method used in Honda 2022. Importantly, even under these conditions, their observation that swimming velocity plateaus at WT levels of flagellar gene expression remains true, further supporting their findings.

With these improvements, the unique contributions of this study have been made much clearer. My take is that whereas the Honda et al. (2022) study demonstrated that bacteria regulate flagellar expression in a manner that keeps the flagellar number constant, the argument for why that specific flagellar expression level is a good choice, namely that it is the 'minimum necessary', remained rather conjectural. The present study makes much more concrete what is meant by the 'minimum necessary' investment - it is the minimum expression level of the regulon needed to achieve the maximum average swimming velocity, and provides a clear demonstration experimentally (by substantially overexpressing the regulon) and theoretically (via RFT calculations; despite its caveats raised by reviewers 5&6, the authors now do a commendable job of demonstrating robustness of the conclusions they draw under different assumptions) that wildtype levels of regulon expression does indeed appear close to the minimum required for maximizing average swimming velocity. Moreover, by demonstrating this using a strain (MG1655) in which motility gene expression has not been artificially engineered, as well as wild isolate strains (ECOR collection), the generality of the argument is strongly enhanced.

Given all of the above, I feel that the manuscript is now much stronger, and I would recommend it for publication.

(Remarks on code availability)

Reviewer #2

(Remarks to the Author)

(Remarks on code availability)

Reviewer #3

(Remarks to the Author)

The authors addressed my questions. I supported the publication of the manuscript.

(Remarks on code availability)

Reviewer #5

(Remarks to the Author)

The two main points we felt needed to be addressed for manuscript to be of sufficient general interest for Nat. Comm, have now been addressed in the revised manuscript.

The first one was on the difference between the Honda et al results and their observations, specifically that their observations hold true for a range of growth conditions, including steady-state exponential growth. Although the authors did not repeat the experiment in the NCM strain used by Honda et al, they show in the revised manuscript that their conclusions hold across a range of growth conditions, including steady-state growth. We think that this shows the results are general enough for wider audience. The authors also discuss the differences between their results and that of Honda et al, specifying that Honda and co-authors did not go above WT level (in fact it seems they could not go above it). While they attribute this to the difference in strains, specifically that NCM3722 was made motile by inserting an IS in front of the flhDC, which is true, we that MG1655 also has non motile WT isolates. Then, the growth rates we observe between these strains (NCM3722 nonmotile and MG1655 motile isolate) are not as big as authors seem to indicate in the response. Certainly, NCM3722 is faster, but not by much. We do however see differences in motile fraction between the two strains, specially when grown on glucose (we describe this further below). This is why we feel that rather than saying MG1655 or NCM3722 are more suited for studying and understanding swimming, a better conclusion might be that how high the expression can go in a given strain, and why, is an interesting question that is yet to be understood.

We also pointed to the RFT model and how it was used compared to previous results. What the authors have presented now is clear, including how/why they picked coefficients different from those of eg Lighthill or Gray & Hancock. This has now been fully clarified and carefully investigated and we believe will be a service to the community. However, the authors might have misinterpreted our comment about the torque under which motors operate (page 16 of the letter). We do not have any issue with the results and overall interpretation that physics limits the maximum swimming speed, as opposed to by under-investment in flagella production. We also agree that motors operate in the low torque regime in those experiments, but we do not think the model can properly estimate the torque actually delivered by individual motors in bundles, because it neglects all filament-filament and filament-body interactions. This is why we suggested removing Extended Fig. 5g, because we are not sure those values actually correspond to the effective torque delivered by the motors, until estimates of internal friction can be obtained. There is a risk that presenting values without experimental backing or a more complete model will ultimately lead to confusion, similarly to the opposite belief that motors of swimming cell operate in the high torque regime, a view that has been prevalent in the literature (especially in theoretical studies) until fairly recently. We think this should be corrected accordingly.

We have a few minor points for authors to consider:

- No values are given for growth rate despite looking specifically at the influence of different growth conditions (and mentioning this in the response to reviewers as an important difference). The competition assays are indeed a better indication of the fitness cost/advantage, but it would still be useful to include some numbers for batch/steady-state cultures and discuss/summarize them. It seems that the authors have measured growth curves for most conditions so this should be straightforward.
- We also struggled to get good motile fractions when growing MG1655 WT in glucose media, while NCM3722 maintains good motility in similar conditions. But in Extended Data Fig. 2: Swimming characteristics in liquid media (well-mixed conditions, no gradients). | Nature, Cramer et al show that MG1655 grown with glucose (triangle at 0.8 h⁻¹ growth rate) has the same speed as in other media. The authors should show the speed and motile fraction separately, so that one can see why the average speed is so low in the current study. It could be because the motile fraction is low while the speed of motile cells is still high?
- L22-23: low average expression levels?

- L48-49: now referring to papers that reported trade-off during experimental evolution, but with same reference list, so not sure it addresses the comment. No evolution in Honda et al?
- L66: we think the present study does not determine those factors either but is showing that cells in rich media are close to the optimum spot on the investment/motility landscape.
- L133-134: data for swimming speed and motile fraction? Would include it in Extended Figure 1 to show why RP437 performs less well than MG1655: lower motile fraction or lower swimming speed or both?
- L138-142: the RpoS deletion experiment and the fact that the authors see no strong difference with the wild type in Fig 1d is interesting. But they only do it in TB, where the expression level of the wild type is already high, and motility near optimal. I don't think this experiment is sufficient to conclude that RpoS doesn't explain the difference between the strains used in this work and the NCM strain used by Honda et al, even if it may very well be true. A more meaningful experiment would be to repeat the experiment in minimal media with the RpoS mutant: this is in those conditions that NCM3722 and MG1655 differ the most. While RpoS levels are indeed low in exponential phase, it seems that E. coli K12 RpoS mutants show higher expression of some motility (MotA/B) and chemotaxis genes during exponential growth, eg Table 2 in <https://link.springer.com/article/10.1007/s00438-007-0311-4#Sec2> (same authors as ref 34-35).

(Remarks on code availability)

We think the results of the paper are reproducible.

Reviewer #6

(Remarks to the Author)

(Remarks on code availability)

REVIEWER COMMENTS

Reviewer #1 (Remarks to the Author):

In this manuscript, the authors titrated the expression of the flagellar gene regulon (placed under the control of an inducible promoter) and demonstrated that the activity of the flhDC (master regulator) operon has a limited physiologically relevant range: the average swimming velocity of the cells plateaus when the flhDC operon activity is induced to wild-type levels and further increasing the operon activity does not increase swimming velocity. Through a physical model, the authors argue that after reaching five flagella — the average number found in wild-type cells — adding more flagella does not increase swimming velocity. The number of flagella depends on flhDC operon activity, indicating that the hydrodynamics of flagellar propulsion limits the physiologically relevant range of flhDC operon activity. Furthermore, the authors tested several wild E. coli isolates and found that when grown on agar, their swimming velocity also plateaus when their flhDC operon activity reaches an activity level close to that of WT MG1655.

This is a compelling result, and the experiments in this paper are very carefully reported. The main suggestion I have is to more clearly delineate the main conclusions of this work from a recent related work by Honda et al. (PNAS, 2022). In the latter study, Honda et al. reported a correlation between swimming speed and PflIA activity, which also saturates at wild-type induction levels (Figure 2 of their paper). Since FliA is a transcription factor that directly controls fliC expression, PflIA activity is closely related to PFLiC activity. Moreover, Honda et al. also demonstrated that the number of flagella depends on PflIA activity, and that motility is maximized when E. coli possesses approximately five flagella (Figure 4, panels E and F of that paper).

We thank the Reviewers 1 and 2 for their careful reading and appreciation of our work, and for the valuable and important suggestion on the improvement of data presentation. We agree with this Reviewer and Reviewer 5 that a more extensive discussion of our results and conclusions in the context of the recent work by Honda et al is important. We now revised the manuscript accordingly, mentioning both similarities and differences between experimental setups and conclusion.

In brief, we believe that there are two key differences in the experimental design between our study and that of Honda et al, which led them to arrive to partly different conclusions. Firstly, their expression system apparently did not allow expression of motility genes beyond the level of their wild-type strain, and the impact of expression on flagellar number and motility above the wild-type level could thus not be explored (their Fig. 4E clearly shows that). Secondly, and more importantly, the strain used by Honda et al, HE204, is not a naturally motile E. coli strain, but it was engineered for motility based on the non-motile strain NCM3722 that was previously extensively studied by that group as a model for proteome allocation. Although this was done by integrating the motility-activating IS element in front of the flhDC promoter in the same way as naturally present in MG1655, such engineered regulation may not necessarily fully recapitulate regulation (including catabolite repression) present in the context of the naturally motile MG1655. For instance, MG1655 largely differs from NCM3722 (and its motile derivative HE204) in its growth rate. This may result in higher basal expression of flagellar genes and motility under catabolite repression (e.g., growth in glucose) in HE204 strain, which would explain the weaker dependence of motility on the carbon source observed by Honda et al. We now discuss these points in the manuscript.

As was suggested by Reviewers 5 and 6, we further performed additional experiments to rule out other potential sources of discrepancy between our studies, demonstrating that it is not due to impact of RpoS or potential difference between batch culture and balanced exponential growth.

Some specific concerns regarding the experiments and various aspects of data presentation (which I believe can be significantly improved) are listed below.

Major:

Reporter Activity Measurements: The authors state that they harvest cells at an OD between 0.4 and 0.6 for TB or 0.3 and 0.5 for M9. However, it is known that tumble bias and swimming velocity of cells (Staropoli and Alon, Biophys. J, 2000) depend on the harvesting OD. Could the rather broad range of harvesting ODs in the present study explain the substantial variability (at least as large as the range of swimming velocity changes predicted by the physical theory) observed between WT replicates reported in Figure 1d?

Although the range of OD₆₀₀ values for which the WT samples were harvested is not that broad (0.48-0.54), this is a valid question and we thank the Reviewer for raising it. We now confirm that, within this narrow range, the swimming velocity does not depend on OD₆₀₀ (new Extended Data Fig. 1e and lines 129-130).

Figure 2d: The swimming velocity predicted by the RFT model only varies between approximately 20 and 24 $\mu\text{m}/\text{sec}$, suggesting the minimum achievable speed with one flagellum is 20 $\mu\text{m}/\text{sec}$. However, the minimum measured swimming velocity is reported to be 5 $\mu\text{m}/\text{sec}$ in Figure 1d. What accounts for this discrepancy? Could the expression of other flhD-dependent genes be relevant? The validity of this physical model would be more convincingly demonstrated if the authors could experimentally verify the conclusions of their RFT model with experiments at the single-cell level. It would seem be technically feasible to assess the swimming velocity of cells whose flagellar filament number are counted by staining with a fluorescent dye.

We apologize for not discussing this point sufficiently in the previous version of the manuscript. As we now explain better in the text, our model does not pretend to capture all details of E. coli swimming (and, to our knowledge, none of the published models can do that). In particular, the assumption of a rigid filament, which is true for a bundle, might be less applicable for cells with a single flagellum, leading to speed overestimation in this case (lines 204-205 and 218-221).

As requested by the Reviewers 3 and 4, we have now extended the analysis of the model, using different assumptions from the literature and different parameter values (see Extended Data Fig. 5, lines 185-217 and Supplementary Note 2). These different versions of the model predict different absolute values of swimming velocity, but all of them captures the general trend of initially increasing velocity and its subsequent saturation with the number of flagella. Furthermore, the values shown on Fig. 1d represent the population average swimming velocity (fraction of swimming cells in the population and their swimming speed are shown separately in Extended Data Fig. 1c and d), whereas the model predicts the swimming velocity of motile cells. To reflect that, we now directly compare the best quantitative model

prediction with the experimental values for the dependence of the velocity of motile cells and the average number of flagella for flagellated cells (i.e., excluding both, the non-motile cell fraction and cells with zero flagella from calculating the data points) (see Fig. 2e). The lowest predicted swimming speeds (20 $\mu\text{m/s}$ for 1 flagellum) still slightly overestimates the speed for the replicates with lowest induction level ($\sim 15 \mu\text{m/s}$), where only swimming speed was measured. Several assumptions of our model likely break down for these replicates and explain the overestimation, such as the assumption of a rigid flagellum for singly flagellated cells, or the value of the flagellar length we assumed being an overestimate for these specific replicates. Measuring cells swimming with fluorescently labeled flagella is possible, but determining the number of flagella in such swimming cells is challenging and error-prone, particularly at higher number of flagella.

Figure 3a: The activity of the *fliC* promoter in WT cells is higher when succinate is used as a carbon source compared to glucose due to catabolite repression. However, this pattern is also observed for the expression of the *flhDC* operon in the inducer-dependent strain. Since the native regulatory region of the *flhDC* operon is absent in this strain, why does it still undergo catabolite repression?

We apologize for not explaining it in sufficient detail. We mentioned already briefly in the text of the previous version of the manuscript that such residual dependence is indeed expected for constitutive (non-catabolite repressed) promoters, due to the global reallocation of protein resources dependent on the growth rate. This was shown, among others, in multiple studies from Terry Hwa's group, and we now provide a relevant citation (lines 229-230).

Minor:

Figure 1b: The x-axis labels are confusing; I recommend using IPTG concentrations instead, similar to what is done for Figure 4b. The WT condition can be indicated with a broken x-axis. Additionally, the y-axis label is simply "fluorescence," whereas in panel c, the mean value of the same quantity is labeled "P*fliC* activity." I suggest the authors reconsider these labels for consistency. The same applies to Figure 3.

We agree and modified the labels accordingly.

Figure 1d: In the figure legend, MG1655 is denoted as "WT", while RP437 and W3110 are not; aren't all these strains wild type for chemotaxis?

*We apologize for the ambiguous labeling. The Reviewer is correct and both RP437 and W3110 are wild type for chemotaxis. We originally specified the WT of MG1655 to distinguish it from the inducible *Ptac* strain that was derived from it. To streamline the notation, we now changed the names of these strains to "RP437 WT" and "W3110 WT" as well.*

Figure 1e: The y-axis label is confusing, "% of WT/*Ptac*" can be interpreted as a ratio of WT and *Ptac* cells, but what the authors are reporting is the fraction of WT or *Ptac* cells compared to ΔFlhC cells.

We agree with the Reviewer and thank for an important comment. The labeling was indeed confusing, so we changed it in the corresponding figure.

Figure 3c: The y-axis label is confusing. What the authors call "positive population" essentially refers to the number of cells with PflIC reporter fluorescence intensity above an arbitrary threshold (as shown in Extended Data Figures 8 and 9). A 100% positive population corresponds to a fully non-bimodal population. I recommend changing the axis label to something like "degree of bimodality" and adding an inset with a cartoon (or actual data) in the white space of this figure to explain how this quantity is defined.

We apologize that our labeling caused so much confusion. We renamed the y-axis as "Positive cells, %" and added an inset with a cartoon illustrating our gating strategy.

Figure 3c: In the figure legend, the IPTG concentration for the Ptac induction is not specified, is it correct to assume that this is at zero induction? This is what line 180 of the text suggests.

Fig. 3c contains the data for all IPTG concentrations for the Ptac strain presented in Fig. 3b, Fig. 1d, Extended Data Fig. 1f, Extended Data Fig. 6b and Extended Data Fig. 9 (including zero IPTG), which explains why the Ptac values vary over the wide range of expression. We thank the Reviewers for raising this point and we now clarify it in the figure legend (lines 910-911).

Figure 4: In multiple axes labels the authors use the notation "(L)" and "(S)" after "PflIC activity" to denote whether promoter activity was measured in liquid or agar. In the rest of the paper, the authors use a similar notation to report whether they used a flow cytometry or plate reader assay. I recommend using consistent notation throughout, for example, "PflIC activity (PR) in Liquid."

We thank the Reviewer for this suggestion. We changed our notation accordingly.

Figure 4b: The x-axis labels after "MG1655 WT" are shifted.

We sincerely apologize for this. The labeling has been corrected.

Extended Data Figure 4: To enhance data presentation, the authors should extract the fluorescence intensities from their western blot and include a plot of the FliC and total protein amount as a function of the induction level. For instance, 3 μ M IPTG seems to induce WT levels of FliC, while in Figures 1b and 1c, it is reported that 10 μ M IPTG induces WT-like levels of PflIC activity.

We would like to point out that this experiment serves only for the illustrative purposes and we are not drawing any quantitative conclusions from it. Reliable quantification of the signals in Western blots is quite challenging and would require additional calibration. Since our main conclusion – that not only the expression but also flagella number can go beyond the wild-type level – is already supported by the direct quantification of flagella number and length (Fig. 2), we believe that the Western blot quantification will not add much to this conclusion.

Reviewer #2 (Remarks to the Author):

We also thank this Reviewer for the positive and helpful feedback on our manuscript.

Reviewer #3 (Remarks to the Author):

The manuscript "Physics and physiology determine strategies of bacterial investment in flagellar motility" by Lisevich et al. reported a study that investigates the relation between the expression of motility genes and the swimming speed and the growth fitness of *E. coli*. The main finding of the manuscript is that hydrodynamics is the limiting factor regulating the motility gene expression. While it is interesting that the authors created a strain of *E. coli* with a titratable expression of the flagellar regulon, the conclusion of the manuscript is neither sufficiently novel nor well supported by the theory. In addition, we found the writing of the manuscript is confusing including experiments and discussions that may not be directly relevant to the main conclusion and theme of the work. Thus, unfortunately, we cannot recommend the publication of the manuscript in *Nat. Commun.*

We listed our comments and concerns below, which we hope can be useful for the authors to revise the manuscript.

We thank the Reviewers 3 and 4 for their critical analysis and feedback on the model and for raising points that require additional clarification and discussion. We would like to emphasize that the modeling was not the main focus of our manuscript, and for that reason it was discussed only briefly. On the other hand, we agree that the modeling part is nevertheless important for drawing our conclusion. We now present a more extensive and detailed analysis as well as description of the model and its comparison with our experimental data (Fig. 2d,e, Extended Data Fig. 5 and Supplementary Note 2) and expand the discussion of our experimental and modeling results in the context of the available literature. Besides, we explain better the connections between individual parts of the manuscript. We believe that now we have addressed all of the criticism raised by these Reviewers.

1) For the fitness measurements (Line 97-104): Although Ptac strains are derived from MG1655 WT, we are wondering if the growth rate of the strains is affected by genetic engineering. A control experiment should be conducted, which compares the growth rate of the Ptac strains with that of MG1655 WT independently without co-culturing. In addition, does the presence of two strains affect their natural growth rates?

The control experiment requested by the Reviewers is already shown in Extended Data Fig. 1a, and the difference between the growth rates dependent on flagellar expression is exactly why this expression imposes growth fitness cost shown in Fig. 1e. Although it is indeed possible to estimate this cost directly from the growth of individual strains, it is less sensitive than the competition between the two strains in the co-culture, which can reveal even minor differences in the growth rates, exactly because strains in the

co-culture compete for nutrients which in turn, affect their relative growth. That is why such competitive growth assays became standard in determining relative fitness of strains/mutants, and to match the definition of fitness in the evolutionary theory. This is a commonly used approach (by us and by others) and we now better explain its rationale and provide a relevant reference in the respective Results section (lines 155-159).

2) As the major conclusion of the manuscript, the authors argued that the increase in the number of flagella does not increase the swimming speed. While this work provides fitness measurements in co-culture as a new way to assess the effect, such a conclusion has already been well established (see e.g. the early discussion by Darnton et al, *J. Bacteriol.* 189, 1756–1764 (2007) and the theoretical/numerical work by Nguyen & Graham, *Phys. Rev. E* 98, 042419 (2018)), which certainly diminishes the novelty of the work.

As mentioned above, theory was not the main focus of our manuscript, and we apologize for describing it too briefly. We would like to emphasize that the key conclusion of our manuscript is not simply that the swimming velocity saturates at high number of filaments (which has been indeed already predicted by previous modeling and discussed as a possible explanation of their results, but not directly shown, by Darnton et al), but that this happens exactly around the experimentally determined number of filaments produced by the wild-type cells. To the best of our knowledge, this was not directly demonstrated previously. We now largely expand the discussion of the model and its findings in the context of existing literature (lines 185-217).

3) Even within the current work, there are a couple of issues regarding the hydrodynamic theory developed by the authors. First, it is well known that the flagellar motor operates with constant torque, instead of constant rotation speed, during the normal run phase of bacteria (e.g. Berg, *E. coli in Motion*). However, the experiments by the authors contradict the established results. How do the authors reconcile the difference?

*As mentioned above, we now describe and discuss the theory in greater detail. We disagree that it is generally established that bacteria operate at the maximal torque when swimming. On the contrary, substantial recent evidence suggests that, in a medium with the viscosity of water like the one we used, bacteria swim in the low torque – high speed region of the torque-speed curve of flagellar motor. Moreover, it was shown that the swimming speed of *E. coli* first stays almost constant when the viscosity of the suspending medium increases (Martinez et al, *PNAS* (2014), Qu and Breuer, *Phys Rev Fluid* (2020)), which can only be explained by the motor operating well below maximum torque in water (probably $\sim 1/3$ of the maximum torque, since a sharp decrease of the speed is observed only for ~ 3 times the viscosity of water). We would also like to highlight that given the experimentally established torque-speed curve of flagellar motor in *E. coli* (see e.g. Chen & Berg, *Nature* (2000)), whether motors operate in the high-torque or the high-speed regime depends on the flagella rotation frequency. The rotation frequency measured in our experiments is 220 Hz, and thus substantially higher than the one measured for example by Darnton et al (possibly because of higher PMF). This likely places our cells already above the transition into the regime with high (nearly constant) speed and low torque (see also the reply to Reviewer 5). Consistently, the data in Fig. 3A of Darnton et al for the fraction of fastest cells indicate that cell swimming velocity may*

saturate already around 200 Hz. We now discuss this in greater detail in the Results section (lines 210-217) and in Supplementary Note 2.

Second, Fig. 1d shows that the swimming speed decreases at the high expression level and exhibits a non-monotonic trend. Can the model also explain the decrease of the swimming speed with the increase of the flagellar number at large flagellar numbers?

As also mentioned in our reply to Reviewer 5, the RFT-based models generally do not capture the drop in velocity at high flagellar number, but always predict a saturation. We suspect this is due to its relative simplicity, notably the assumption of a tight bundle, which is unlikely to hold perfectly at very large flagella number. However, what is critical to our argument is that an increase in flagellar number does not increase swimming speed, something that all tested models capture well. We now comment on these points in greater detail in the Results (lines 193-209), and further in the additional discussion of Supplementary note 2.

3) Indeed, it is an open question why bacteria grow multiple flagella, as the number of flagella does not increase the swimming speed of bacteria. Many other possible functional benefits have been discussed (e.g. the increase of swimming stability or the enhancement of surface attachment). We would suggest the authors to survey the extensive literature on the topic more thoroughly. It is hardly convincing based on a single observation and a simplified (maybe questionable) calculation to conclude that the hydrodynamics is the sole factor limiting the growth of more flagella.

*What we show is exactly the opposite: that wild-type *E. coli* does not synthesize more flagella than necessary to achieve maximal velocity. We would like to emphasize that this conclusion is based not only on the theory (and we expanded the analysis of the model, to show that its general conclusions do not depend on the details) but more importantly on direct experimental measurements for multiple *E. coli* strains, including not only laboratory reference strains but also natural isolates. We thus do not see the necessity to invoke other, more speculative explanations. As a side note, an increased stability of swimming with multiple flagella, e.g. by reducing the wobbling of the body, might contribute to the experimentally measured increase in the swimming velocity, as mentioned by the Reviewer. Our revised model accounts for cell body wobbling, and we now show the predicted dependence of the tilt angle on the number of flagella (Extended Data Fig. 5g), which indeed decreases in our model. However, the predicted dependence of swimming on this angle is weak (Extended Data Fig. 5h) and it does not change our general conclusion about the physiological reasons for saturation of motility with flagellar gene expression.*

4) The authors showed that the motility gene expression affects both the number and the length of flagella. We are wondering if other properties of flagella (e.g. pitch angle, helical radius et al) are also affected and how they impact the observed behavior.

We would not expect the expression to change the helical properties of the filament, given that flagellar filament is a regular structure assembled from subunits of a single protein.

5) The authors argued that "native gene expression under carbon-limited growth might correlate with the

potential benefit that could be achieved in a given carbon source by performing chemotaxis towards sources of additional nutrients". It is far from clear how the expression of motility genes (or the number of flagella) affects chemotaxis. It seems that the authors tried to distinguish swimming speed and chemotaxis as two independent effects. Maximizing one does not affect the other. In fact, the swimming speed and chemotaxis are closely related. A lower swimming speed leads to poor chemotaxis. After all, swimming evolved to achieve chemotaxis.

We fully agree that swimming velocity and chemotaxis are tightly related, which is well established. The full sentence quoted by the Reviewer(s) reads "we hypothesized that native gene expression under carbon-limited growth might correlate with the potential benefit that could be achieved in a given carbon source by performing chemotaxis towards sources of additional nutrients, as proposed before"²¹. It refers to our previous publication Ni et al. PNAS, 2020 that showed that the expression of flagellar genes (and hence both swimming velocity and chemotaxis) in cells growing on different carbon sources correlates with the benefit of chemotaxis under these conditions. We now modified this sentence to make it clearer (lines 243-247).

6) It is not clear how the two subpopulations are related to the main conclusion of the manuscript, i.e., hydrodynamics determines the motility gene expression. Some clarification of the relation between the two discussions is necessary, otherwise the discussion is a distraction of the main theme of the paper. Moreover, the authors argued that "the negative regulation of FlhDC activity by YdiV" is not the origin of bimodality. Then what is the origin?

We now describe and explain this section more extensively, including giving a separate title (lines 257-294). In a nutshell, our interpretation is that producing less than one to two full-length flagella on average becomes physiologically meaningless because it would not allow efficient swimming but nevertheless impose the cost. As for the potential origin of bimodality, we performed additional experiments (now shown in Extended Data Fig.9 and in Fig. 3c) that demonstrate the involvement of another negative regulator of flagellar expression, FlgM. We thus argue that both YdiV and FlgM contribute to the bimodal expression of flagellar genes, with FlgM having a larger impact, although further research going beyond the scope of this manuscript is needed to reveal the mechanistic details of this regulation.

7) Swimming in bulk fluid and swarming on agar surface or in porous media are quite different from the hydrodynamic standpoint of view (see e.g. Kearns, Nat. Rev. Microbiology 8, 634–644 (2010)). So, the discussion relating the locomotion of bacteria in the bulk fluid and along the surface of agar is quite confusing. It is not clear if the conclusion reached by the swimming in the bulk fluid via hydrodynamics can be translated to the swarming in agar plate. More evidence is needed to support the hypothesis on "some deficiency in flagellar assembly or function in liquid-grown cell".

We apologize, but this is a misunderstanding: we have not investigated swarming here. For E. coli, swarming on a surface requires very specific conditions of incubation, with 0.45% special Eiken agar, which is not what we did. Spreading in 0.27% soft agar tested in our work means swimming in a porous medium, not swarming on a surface. We also grew E. coli the 0.5% surface of regular agar but subsequently tested their swimming in the liquid, and again not locomotion along the surface of agar. We now emphasize this difference more explicitly in the Discussion (lines 394-399) and cite the reference mentioned by the Reviewer. But we also mention (lines 398-399) that the two phenomena might be potentially related.

8) While it is reasonable that different niches and environments lead to different saturated numbers of flagella, why does the MG1655 WT provide the upper limit of all different strains?

We have not concluded that “different niches and environments lead to different saturated numbers of flagella”. What we conclude is that the selection for motility may depend on the environmental niche, whereas the physics of swimming remains the same. MG1655 is at the upper limit of the number of flagella exactly because a further increase in this number will not increase swimming velocity, neither for MG1655 nor for natural isolates. Other isolates may have optimized the trade-off between motility and growth to different points of flagellar gene expression, even under the same limitations described here, because of weaker selection for motility and/or stronger selection for growth in their respective niche. We now further elaborate on this interpretation to avoid misunderstanding (lines 365-380).

Minor points:

1) In Line 43, the authors stated, "Motility consumes several percent of total cellular resources ..." Could the authors be more specific and clarify the range of the value?

Since the exact fraction of cellular resources consumed by motility depends on the expression level of flagellar genes (what is studied here), there is no single number that can be given. But we now mention the range of 2-8% determined by Basan et al., 2015, which also agrees with other mentioned estimates (and is reflected by “several percent” used in the previous version of the manuscript) (line 47).

2) In Line 91, the authors claimed that the two other derivatives of E. coli showed a similar relation. Nevertheless, from Fig. 1d, the two derivatives have fixed gene expression without showing the non-monotonic trend. So it is unclear what the similar relation means here.

We apologize if this was not clear, but for these two derivatives we only tested the wild-type strains (which we better emphasize now by labeling them as WT, as suggested by the Reviewer 1), not the inducible derivative as was done for MG1655. We also rephrased this particular sentence to say that they map to the same curve (lines 132-134).

Reviewer #4 (Remarks to the Author):

We also thank this Reviewer for feedback and suggestions on improvement of the presentation and discussion of our results.

Reviewer #5 (Remarks to the Author):

The authors study motility regulation in E.coli, but because they do not put their work well in the context of the existing literature the manuscript is difficult to follow. As far as we understand they build on the work from Honda et al PNAS (<https://www.pnas.org/doi/10.1073/pnas.2110342119>) that shows exponentially grown E.coli upregulated gene expression in nutrient-poor environments to maintain the same flagella number per cell because the cell size decreases. The swimming speed and motile fraction thus stay roughly constant across growth conditions (i.e. both nutrient-poor and rich growth media). In the Honda et al PNAS study authors titrate the expression of flagellar genes and show that the maximum expression they can achieve corresponds to the level observed in WT and to 4-5 flagella per cell, where roughly that number is needed to achieve wild-type, exponential growth motile fraction and swimming speed. Here authors achieve expression above the WT level and show that at that level of expression, although the cells make more filaments, they do not swim faster, limited by the hydrodynamics of swimming.

For this result to be of interest to the general audience of Nature Communication the authors would need to demonstrate that what they observe is a general observation.

We thank this Reviewer and the Reviewer 6 for their very thorough reading and extensive and helpful feedback on our manuscript. We now discuss the differences between our work and that by Honda et al in a much greater detail, indeed arguing that our conclusions are likely to be more general whereas those by Honda et al may be limited by the specific strain that was used in that study. Please see our responses to specific points below, as well as our responses to Reviewer 1, who has raised similar points.

To start, why is there a difference between the authors' observations and those of Honda et al mentioned above?

One explanation would be that Honda et al simply did not induce above the WT level so have never seen that expression goes further up, this seems unlikely though (is it?).

We believe that this is indeed the case. Their Fig. 4E clearly shows that they only titrate the number of flagella and swimming velocity below the WT level. This seems to come from the limitation of their expression system, because it is the expression level that saturates with the concentration of inducer (and always remains below WT) and not the swimming velocity or the fraction of motile cells (compare their Figs. 2A with 2C,D and also see their Fig. S4). We now mention it (lines 63-65 and lines 350-352).

Other differences between this manuscript and previous work are the strains (here MG1655, and in Honda et al NCM strains) and the stage of growth (exponential phase in Honda et al, and undefined state here). The authors would need to perform experiments that tell us what the source of the difference is. To check if it is the strain authors would need to compare NCM and MG1655 strains in exponential phase. If it is a strain difference, and it is known that NCM has an issue with RpoS authors should then check if NCM strain repaired for RpoS behaves as they see for MG1655, all in exponential phase. If this is the case, their results are of interest to a general audience of Nature Communications, because this would indicate that what

they observe here is more what we should expect from wild-type strain than what Honda et al observed, with NCM strains defective in RpoS.

If it is not, if it is simply the stage of growth, then this result is more suited for a specialised audience journal, as it depicts what they see when they grow the cells in the specific way that they do. And that state is not well defined.

To expand on the stage of growth. Here the authors are growing the cells in Tanaka media with defined carbon sources, as well as in TB and LB where cells are consuming one preferred amino acid after the other in series, and as soon as they run out of the most preferred one. They also dilute their culture o/n, either 1x100 or 1x1000, which is not enough for cells to get to the exponential phase (they would have not fully diluted from the o/n stationary phase in the little generations they can achieve without running out of some nutrient). In Honda et al, cells are heavily diluted at low ODs to ensure they do not run out of any nutrient and reach at least 6 (but most of the time more) generations in such a constant environment. Not to say that studying other stages of growth is not interesting, in fact, bacteria are hardly ever in such a well-defined exponential phase. The problem is that the state authors have them in is not well defined. The exponential phase means cell number, biomass, and even individual cells are all growing at that exponential rate, with the previous phase well diluted. We know from 'Copenhagen school', i.e. the older papers from Schaechter, Maaløe, Kjeldgaard, etc that the way cells exit the exp phase (i.e. what changes in physiology first) depends on what has run out in the media. Leaving us with interesting, but not well-defined growth phases. And, what happens there is at this stage of more interest to the specialist audience (until we find a way of better defining those growth phases).

We thank the Reviewer for this very careful and thoughtful considerations of possible differences between these two studies. As also elaborated on in our response to the Reviewer 1, we believe that the crucial difference is indeed in the strain HE204 used by Honda et al (the first possibility stated by the Reviewer). NCM3722 is an excellent and a well-studied model for the growth-dependent proteome resource allocation, but it is originally non-motile. Its motile derivative HE204 (parental of NCM3722B mentioned by the Reviewer) used by Honda et al was engineered in a previous study by Cremer et al 2019 by integrating the motility-activating IS element in front of the flhDC promoter in the same way as it is naturally present in MG1655. While this was arguably the best way to activate motility of this model strain, and fully suitable for the original study by Cremer et al, such artificially engineered regulation may not necessarily fully recapitulate the details of the evolved regulation (including catabolite repression) present in the slightly different physiological context of the naturally motile MG1655 (which e.g. differs from the NCM strains in its growth rate). We believe that it simply results in higher (basal) flagellar expression and motility under catabolite repression (e.g., growth in glucose) in HE204 strain. Together with the aforementioned limitation of their induction system, this can largely explain the differences in our conclusions. This is now mentioned in the Discussion (lines 405-415).

Although we could indeed engineer and study such motile version of the NCM, we honestly do not believe that it is necessary. We do not question at all the experimental findings of Honda et al (and thus do not see a need to verify them), but would simply argue that this strain is not the best benchmark model to study evolutionary optimized regulation of motility, given that motility was never exposed to the evolutionary selection in this specific background.

To add to that, we also do not question that increased flagellar expression can compensate for the decrease in the cell size in poor carbon sources, as concluded by Honda et al. What we say is that this is

not the whole story and alone it cannot explain the more pronounced catabolite-dependent regulation of flagellar expression observed in the naturally motile MG1655. We now specifically mention in the Discussion that these interpretations are not mutually exclusive (lines 414-417).

In order to further address possible impact of RpoS, which is non-functional in NCM, we measured the expression and swimming in the $\Delta rpoS$ knockout constructed in the background of our parental MG1655 strain. We observed no major difference between the strains (Fig. 1d), indicating that RpoS has little impact on the observed regulation of motility, possibly because the RpoS is largely inactive in the mid-log growth phase (lines 137-142).

Finally, since the Reviewer also raised a valid point about possible differences between growth conditions (batch growth in our study vs balanced exponential growth used by Honda et al), we performed additional measurements under balanced growth, both in TB medium (Extended Data Fig. 1f) and in the minimal medium (Fig. 3b). Also in this case, the relative differences in expression and swimming between MG1655 WT grown on glucose and succinate were apparent and similar to our original measurements in the batch culture. We therefore conclude that our results are similarly valid under conditions of balanced growth (lines 239-241). We thank the Reviewer for suggesting these additional experiments, which we believe were very helpful to further generalize our conclusions.

The authors complement their experimental findings with theoretical predictions based on the Resistive Force Theory, which we found well-written (and service to the community to have the derivation as well explained as here), but also found some mistakes. We believe these are important because it will ensure authors put less emphasis on 'excellent agreement' between RFT and their data. Once we corrected the found problems we did not see such a good agreement. For their arguments, this is not important, but it is for estimates on how good of an approximation RFT is (not as good as authors seem to argue).

We thank the Reviewers for their appreciation and careful inspection of our model. We now substantially revised and revisited the model, including surveying the predictions of several competing models from the literature, and our own variations around these, for the flagellar friction coefficients that the referee did not find satisfyingly modeled. While some models are quantitatively better than others, they all show the same qualitative behavior as the experimental data. Most importantly, the lack of swimming speed improvement at higher flagella number, which is the crux of our experiments, is captured by all models. This supports our claim that the physics of swimming sets the upper bound on swimming speed we observe, with which we believe the Reviewers agree. As for the quantitative agreement with the data, it indeed depends on the version of the model and on the exact expressions for γ_k and γ_t that are not unambiguously defined. We thus used expressions that allowed us to reproduce the experimental data very well. But we absolutely agree that this does not mean that the RTF model provides a perfect quantitative description of bacterial swimming with multiple flagella. We improved the description of our modeling approach in the main text and in the Supplementary Note 2, and also clearly state that the model is simplified. However, as also mentioned by the Reviewers, what is important for making our argument, is the robustness of the qualitative prediction of the model, which is now emphasized in the text when discussing the modeling results (lines 193-209).

Following are some comments on specific lines in the manuscript (including minor typos etc) for authors to consider:

We thank the Reviewer for this detailed highlighting of specific points that could be improved.

Title -suggest to change. Currently, it does not mean much

We modified the title to make it more specific.

The abstract is not very clear and we suggest authors make it punchier and easier to understand. Line 14, costly needs to be put in context. It's cheap to power flagellar motility from the point of view of cellular energetics, but expensive to make it from the point of view of protein allocation
Line 15-17 it's a bit of an awkward phrasing, suggest revising the sentence
Line 18-19 it's not obvious what the authors mean by 'physiologically relevant range of investment' before reading the manuscript. It would be useful to revise. The rest is unclear as well
Line 12 a typo 'for different functions' rather than in (although happy for authors to leave this to be corrected by the journal staff and given the publication charges). We would not agree with 'poorly' either as there has been a decent amount of work on this recently, we'd simply say that it is not fully understood yet.

The Abstract was modified accordingly (lines 11-25).

Line 30 typo, capital T at the beginning of the sentence (as above when it comes to typos)

Corrected.

Line 44, some of the key references for this are missing, such as

<https://pubmed.ncbi.nlm.nih.gov/26632588/> (protein reallocation)

<https://pubmed.ncbi.nlm.nih.gov/21390082/>

<https://pubmed.ncbi.nlm.nih.gov/17438023/> (growth curves with and without FlhDC)

We cited these additional references, and thank the Reviewer for suggesting them.

Line 45, there is a key figure for this claim in ref <https://pubmed.ncbi.nlm.nih.gov/26632588/> as well

We thank the Reviewer for pointing us to this figure. We agree, and now refer to this study in order to give a more specific number as asked by the Reviewer 3 (line 47).

Line 45-48 This is confusing and is citing papers that claim different things as if they claim the same (to our understanding of these papers). For example, the following paper <https://www.pnas.org/doi/full/10.1073/pnas.1910849117> by some of the authors, claims that

increased gene expression in poor conditions gives an advantage because the cells can follow nutrient gradients in those conditions (i.e. have a chemotactic advantage because the swimming speed and chemotactic drift velocity increase). Whereas this publication <https://www.pnas.org/doi/10.1073/pnas.2110342119> says that an increase in gene expression in poor nutrient conditions is there simply to have a constant flagella number as cells become smaller. And, that cells' swimming speed and ability to do chemotaxis is not dependent on the growth rate so that cells have that swim and chemotax in nutrient-rich conditions expand faster than those in poor <https://pubmed.ncbi.nlm.nih.gov/31695195/>. So to group it all under one as 'trade-offs' is misleading.

We modified this part of the text to specifically refer to the papers that reported such trade-off during experimental evolution (lines 48-49).

Line 49-56 This now explains better my comment above. But I am leaving it, as that information was needed before. I think the authors should do a better job of explaining this straight away (in fact maybe those last sentences, lines 45-48 are not needed, and instead this paragraph should simply follow, where the difference in explanations is made clearer).

We modified this paragraph by removing the last sentence, which we agree was not necessary (lines 49-50).

Line 57-58 But this has been done in all those publications we are discussing above, and it leads to opposing opinions. So how is what is being done now different? And what is the main reason these experiments are now done? What exactly do the authors wish to investigate given what is given in previous literature? All this should be made clear.

Line 59-64 It is not really clear from this what the authors learn new, as opposed to what is in agreement with previous literature.

We modified and largely expanded this last part of the Introduction to explain better the rational of our investigation and where our findings go beyond what is known from previous studies (lines 52-66).

Line 69 and the paragraph that follows. In light of the previous literature (for example, figures 1 of both of the above PNAS papers) it is not clear why the authors are doing this? And how the results are similar and different from what has been done before. Please put the results in a better context of previous literature so that the novelty of the current results is clear.

We modified this paragraph to explain better the rational for our measurements and their differences from previous studies (lines 67-71).

The authors also mention different strains and include them. But, as mentioned above the most relevant previous publication for this work uses NCM strains (and those can have an issue with RpoS), and that has not been included.

*As discussed above, we would argue that remeasuring NCM strains is not necessary, and we have now confirmed that deletion of *rpoS* has only minor if any effect on our results. We also added the measurements for cultured under balanced growth (Extended Data Fig. 1f and Fig. 3b), which confirm our results for the batch cultures (lines 143-150 and 239-241).*

It is great that the authors are using RP437 (and W3110), because that strain is frequently used for swimming as they say. But there are hardly any other physiological studies on it, without doing all of those, this comparison here is not as informative.

*The main reason for measuring these strains, as well as with natural *E. coli* isolates, was to demonstrate that swimming scales with expression in a strain- or growth conditions-independent way, and to test where do the other naturally motile derivatives of K-12 fall on this dependence curve. We believe that this comparison will also be useful for people using these strains for motility studies, particularly in case of RP437. The fact that RP437 is suboptimally motile further helps to explain why this strain could be experimentally evolved to enhance its motility in our previous study Ni et al, 2017 (and we mention this in the text, lines 132-137). Such evolution should not be possible for MG1655, since its motility is already at the maximum. It was indeed not possible when we tried it before (our unpublished data).*

Line 75: double "in" typo.

We sincerely apologize for our typos and thank the Reviewer for noticing them. The typo was corrected.

Line 106 and corresponding results chapter. While the authors describe their results well, these have not been put in the context of existing literature. For example, <https://www.pnas.org/doi/10.1073/pnas.2110342119> Figure 4E or S

This results chapter has now been revised to discuss our finding in the context of the work by Honda et al. The figure to which the Reviewer refers actually demonstrates that this study only explored expression levels below those in the WT strain, and it could not draw conclusions about the impact of expression and flagellation above the WT level (and thus about saturation of motility with expression). This is now mentioned (lines 63-65 and 350-352).

Line 122-123 We disagree with this statement. The flagella are not at the pole like in RFT, but randomly distributed along the cell body. Thus, the torque the motor is producing is perpendicular to the length of the swimming. It is not at all clear therefore that this approximation is the best for understanding the hydrodynamics of *E. coli* swimming. It seems to be simply what we can do now. Line 124-125 please add a bit more details on the assumption (e.g., and importantly, all the motors are at the back of the cell body)

Note that in our previous force-torque balance analysis (FTBA), we did not assume a specific location for the flagella motors. We now expanded the model to account explicitly for the position of the motors on the cell body. We also corrected a previous (small) mistake, which changed the expression of the rotational friction coefficient, with however little change to its value. We find that most of the results of FTBA are independent of the position of the motors on the cell body. In fact, only the inclination (θ) of the cell

body relative to the axis of motion depends on motor positions in our extended analysis. As we now discuss in the text (lines 193-209) and in Supplementary Note 2, the model is still not perfect and clearly simplified, but we believe it is sufficient for robustly drawing conclusions in this paper (lines 218-221).

Line 125-130 We think that the fact the authors see that the motor speed is independent of the flagella number is a result in itself and the authors should put more emphasis on it. But, we disagree with the interpretation that follows the description of this result. The zero-torque speed of the E. coli motor is reported close to or higher than 300 Hz (<https://www.sciencedirect.com/science/article/pii/S0006349500766628>, or <https://www.ncbi.nlm.nih.gov/pmc/articles/PMC5703321/> or) at room temperature in the presence of a good carbon source. 220 Hz is not far from the 'knee' (<https://www.sciencedirect.com/science/article/pii/S0006349500766628>) of the torque speed curve and not much faster than the rotation speed of a 0.5µm bead in those conditions, i.e. not zero-load ([https://www.cell.com/biophysj/pdf/S0006-3495\(19\)30392-3.pdf](https://www.cell.com/biophysj/pdf/S0006-3495(19)30392-3.pdf)). Thus, the motor speed should change if the load changes. The fact that it doesn't with an increasing number of flagella suggests that the torque supplied by each motor is roughly constant, and thus that the increase in load due to larger flagellar length and bundle thickness is balanced by the fact that more motors share that load, hence placing each motor under a roughly constant load as a function of N. We noticed that the authors did mention this as a side note, in Line 137-139 of the SI. We think it's a major contributor.

The Reviewer is right in raising this point that we agree was not sufficiently discussed in the previous version. However, we would like to emphasize that the value of the maximum speed of ~300 Hz (knee ~200 Hz) corresponds to a temperature of 22.7°C in the Chen2000 paper (their Fig. 5), whereas our experimental room temperature was 20.5±0.5°C. For a temperature of 17.7°C, Chen and Berg found a maximum speed of about 170-200 Hz with a knee slightly above 100 Hz. A linear interpolation between these two temperatures suggests that 220 Hz may not be far from the maximum speed at our temperature. We also have to point out that we are not using the same strain as Chen & Berg, and that there could easily be additional (small) difference from strain to strain in the maximum speed at a given temperature. Moreover, cells operating close to the high speed limit is consistent with the fact that swimming speed is initially almost constant when increasing suspension viscosity, e.g. with ficoll (Martinez et al, PNAS (2014), Qu & Breuer, Phys Rev Fluids (2020)). We therefore think that the motor speed being at (or close to) the limit is a more parsimonious explanation at this point. Nonetheless, we accept that there might be some additional mechanisms fine-tuning the motor speed to remain constant but somewhat below its maximum, but figuring if and how this happens is far beyond the scope of this paper. This is now discussed in the text (lines 210-217) and in Supplementary Note 2. We agree with the referee that the constant motor speed is an interesting observation in itself, and we include it in the main figure (Fig. 2d).

Linked to this, we suggest removing Figure 5g unless it can be proven experimentally because it is formulated for a single perfect helix (i.e. perfect, tight bundle) and neglects all possible interactions between multiple filaments.

The purpose of former Extended Data Fig. 5g (now replotted as Extended Data Fig. 5i) is to illustrate the implications of our model, and to test its internal self-consistency. Because we model the motor speed to be constant, we effectively assume to be in the high speed-low torque region of the motor torque-speed

relationship and it is important to confirm that the speeds and therefore the torques our model predicts are consistent with this assumption. Similarly, we assume theta to be small, which we now check in Extended Data Fig. 5g. This is not central to our argument of speed saturation, but this is typically the kind of internal check that should be shown in supplementary, in our opinion.

Furthermore, by rearranging $\omega_m = \omega_f(N) - \omega_b(N) = cst$, and $\Gamma_f = \mu_{RR} \omega_f(N) - \mu_{TR} U(N) = -\Gamma_b = -\mu_{RR} \omega_b(N)$, one obtains the decay of bundle rotation speeds seen in ED Fig. 5c (although with slightly higher frequencies see note below), thus the assumption that the motor torque decreases roughly as $1/N$ isn't needed to explain the experimental results.

The Reviewer is correct. This is precisely the reason why our theory explicitly uses $\Omega_M = cst$ to derive speeds. The motor torque decreasing roughly as $1/N$ is an outcome of the model, not an assumption. The friction torque on the flagellar bundle, which is equal to the one on the cell body because of torque balance, being shared between all the motors induces the $1/N$ dependence in our model. We reformulated Supplementary note 2 and the main text (lines 197-200) to make this clearer.

Line 130-132 Why not include the experimental data in Fig. 2d? Please also cite previous literature and acknowledge that it shows the same with slightly different parameters <https://www.sciencedirect.com/science/article/pii/S0006349595800895?via=ihub>, (Fig.7b)

We now include the experimental data in Fig. 2e (former Fig. 2d). Although a beautiful pioneering work, Magariyama et al (1995) does not show quite the same thing as Fig. 2d, since they studied V. alginolyticus that has only one filament. We now mention in the main text (lines 203-204) that it predicts a similar saturation of speed as a function of flagellar length.

Line 131-138 While the model predictions are in agreement with experiments authors put a too strong emphasis on this given that they have made a mistake in the ratio of the drag coefficients (in SI calling it 'the agreement is overall excellent'). As a result, they think RFT is overall a better predictor of E.coli swimming speeds than it might be (we think it's simply indicative). The γ_k parameter in Eq 2.16 of the supplementary note is the ratio of the tangential to normal drag coefficients in Lighthill's analysis. It is not constant when the thickness of the effective filament varies. Furthermore, the value of 0.7 was obtained for sea urchin spermatozoa, not bacteria. Using the coefficients derived by Lighthill in Flagellar Hydrodynamics (1976), $K_n = 4\pi\eta / (\ln(0.18\Lambda/r_f) + 0.5)$ and $K_t = 2\pi\eta / \ln(0.18\Lambda/r_f)$ with $\Lambda = \lambda / \cos\psi$, we obtain significantly higher swimming speeds than the ones predicted in the manuscript. Using a coefficient of 0.25 (as in the manuscript, not clear why?) instead of 0.18 (as above) doesn't affect this. Authors should update the predictions with the correct parameters. We note that RFT obtained speeds larger than those observed in experiments would not be surprising, since RFT doesn't account for hydrodynamic interactions. It would then also be good to get the flagella rotation speed from the model for a given motor speed in ED Fig. 5c?

We thank the Reviewer for the critical detailed study of our model. When building the model, we tried several expressions for the coefficients k_n and k_t , including Gray & Hancock's RFT, Lighthill expressions,

and effective models where we fixed γ_k , inspired in part by Chattopadhyay2006. All models predicted the same qualitative behavior, but given the known shortcomings of RFT-based models, as the referee points out, Gray & Hancock and Lighthill original expressions were not good quantitative predictors. We thus settled on an expression close to Lighthill with a fixed γ_k . We now explicit this approach in supplementary material, showing the results for the analytical and effective models for k_n and $\gamma_k = k_t/k_n$ in the revised ED Fig 5. We find that the quantitative predictions of Lighthill-like models are extremely sensitive to the value of γ_k , because of the $(1-\gamma_k)$ term in the thrust coefficient μ_{TR}^f . We note that bringing γ_k to a value close to 0.7 is a very effective way to bring Lighthill-like models, which are otherwise strongly overestimating speeds, into very good agreement with the experimental data. Finally, the Lighthill-like model with k_n from Lighthill and $\gamma_k = 0.7$ is almost identical to using $c=0.25$ and $\gamma_k = 0.7$ as we did in the previous version of the manuscript. We thus decided to show the results from the former, and to remove the coefficient $c=0.25$ from the text for simplicity. Furthermore, we are now emphasizing in the text (lines 193-197) that our key qualitative conclusion (velocity saturation at high flagellar number) is valid for all the tested models of the friction coefficients of a helix.

It also seems that the authors consider a diameter of 40 nm for an individual flagellar filament, but we think values around 20 nm are more common, suggest to double check.

We indeed mistakenly used the diameter value for the radius, and we thank the Reviewers for pointing it out. We corrected this error, which however had a negligible effect on the speed predictions ($\leq 5\%$).

Line 143-144 and the corresponding results chapter. Here again, we feel the results are well described and need to be put in better context with previous literature. The result of Honda et al in PNAS paper Figure 2C is similar to figure 3b here, however, there cells grown in different carbon sources reach the same speed that corresponds to the maximum gene expression level. Here, they don't, and the average swimming speeds are lower for WT in media with lower nutritious value. This is consistent with the following results by some of the authors <https://www.pnas.org/doi/10.1073/pnas.1910849117> (specifically one of the SI figures there) But here again, the main difference seems to be the way the cells are grown. Following previously published figure <https://www.nature.com/articles/s41586-019-1733-y/figures/6> makes this explicit. So we are not sure what these results add? If the cells are grown in the particular way the authors are growing them, and that is not a well defined state, the result is the same across many different media? This could be interesting as it indicates that the way authors are growing the cells is a 'state' in the physics meaning of the word. But if that is the point this section needs rewriting, and also plotting in a slightly different manner (maybe all the swimming speeds obtained in different media when cells are grown like the authors grow them need to be plotted against the growth rate at the time the cells were collected).

As mentioned above, we now confirm our conclusions under conditions of balance growth, suggesting that they are not specific to the particular way our cultures are grown but rather general. These results are now discussed in the manuscript (lines 239-241).

Line 154-159 Yes, but it disagrees with the other two studies as mentioned above and the authors are not adding anything to explain the disagreement.

As discussed above, we believe that the disagreement is due to the difference between strains, which is now mentioned in the text (lines 405-415).

Line 166-168 Or it is simply a consequence of not properly diluting yet from o/n growth. Needs checking whether the bimodality would still be there if cells were grown in exp phase. Otherwise, it's just an observation for the cells that are grown in a particular way of growing cells.

Line 174-177 but neither seems enough. Can the authors calculate how many generations the cells achieve with 1x100 and 1x1000 dilution at their growth rate to the OD they harvest them at? We suspect $10^4/10^5$ dilution will be needed to get 6-10 generations.

Usually a TB-grown overnight culture reaches $OD_{600} \approx 2-2.5$. Hence, our starting $OD_{600} \approx 0.02$, so that cells need 4-5 generations to reach $OD_{600} \approx 0.4-0.6$ if diluted 1:100 and 7-8 generations if diluted 1:1000. The dependencies of the fraction of bimodal cells on the expression level falls on the same curve in both cases. We believe that this rules out the incomplete dilution as the cause of (highly pronounced) bimodality in our experiments. We mention it now in the manuscript (lines 274-275).

Line 180: ref to Fig.1b, not 1c.

We sincerely apologize for this typo. The typo is corrected.

Line 196 and corresponding results section. This is potentially very interesting, but it's hard to interpret because cells were again not grown to the exponential phase (and the way they were grown could be a different state for each different isolate). It either needs growing into exponential phase or a better understanding of the current state the cells are grown in. Motile fractions and the swimming speeds of motile cells should also be provided for those experiments. At the moment it is unclear if it is the motile fraction or the speed of the motile cells or both that change when cells are grown first on semi-solid medium.

Although the majority of ECOR strains grow differently (mostly faster) compared to MG1655 WT (Extended Data Fig. 10a) they show a similar kinetics of activation of flagellar genes (Extended Data Fig. 10b) justifying our choice of the particular OD_{600} range to do the measurements. We have now added the fractions and swimming speeds of motile cells (Extended Data Fig. 10). These data demonstrate that, for the majority of ECOR strains, swimming fractions change much stronger than the velocities of motile cells when incubated on semi-solid medium. Both these points are now discussed in the text (lines 307-311 and 331-333).

Line 335-337: everything should be written in mM for the minimal media to allow comparisons between the two. Also, in the original paper Tanaka medium contains 20 mM $(NH_4)_2SO_4$, 34 mM of NaH_2PO_4 and 64 mM of K_2HPO_4 . Typo or modified medium?

We apologize for the typo in Tanaka medium receipt and for inconsistency in units. All the corrections were made (lines 464-471).

Line 386 that is a very small volume for DDM experiments. What is the thickness of the sample, and how far away from the glass slide are the cells imaged? Can the authors exclude wall effects?

We sandwich a cell suspension drop between two glass coverslips, which are held separate by grease. The drop is $\sim 500 \mu\text{m}$ high and $\sim 3 \text{ mm}$ in diameter (for $3 \mu\text{L}$ volume). We stay at least $100 \mu\text{m}$ away from all edges, more than the minimum $50 \mu\text{m}$ standard for DDM measurements. Therefore, their effect is as negligible as possible, given the long-range nature of hydrodynamic interactions. This is now clarified in the Methods (lines 339-341).

Supplementary Note Line 93-94. Yes for the thrust, but in ref 10 they show a major drop in hydrodynamic efficiency if helices are out of phase, because the drag on resulting propeller is increased and thus the speed decreases. So, the bundle configuration does matter.

This part of the Supplementary Note was rephrased and the sentence removed. In former ref 10 (now ref. 11) of the Supplementary Note, thrust is unaffected and rotational drag seems to increase only moderately (judging by the moderate increase in dissipation). They indeed observe a sizeable decrease in induced flow field (factor ~ 2 , from the hydrodynamic efficiency) for opposite-phase helices, which should correspond to an equivalent increase in translational drag. This increase supports the argument that the bundle possibly getting less tight at high N could contribute to the speed reduction that our model does not capture. This is now mentioned in the additional discussion of Supplementary Note 2.

Supplementary note, RFT section: typo when || is used instead of index b for body coefficients?

We thank the Reviewer for pointing out this typo. It is now corrected.

Reviewer #6 (Remarks to the Author):

We thank this Reviewer for the thorough and critical reading of our manuscript and for very helpful comments for its improvement.

Reviewer #1 (Remarks to the Author):

The authors have addressed all of my requests from the previous round, as well as many comments/suggestions from the other reviewers. As a result, the manuscript is now substantially improved.

Most notably, they now much more clearly explain the difference between the present study and that of Honda et al. (2022, PNAS). Whereas the Honda paper only titrated expression of the flagellar regulon below WT levels, the present study used an expression system capable of increasing the expression of motility genes above WT levels, to clearly confirm that motility levels saturate very close to the wildtype expression level of the regulon. Additionally, they now clarify that the Honda et al. study used a strain (HE204, derived from NCM3722) that is not naturally motile and hence has not been evolutionarily optimized for motility. This distinction is significant in considering the generality of the results, which are further strengthened by additional experiments with wild isolate strains. The revised text emphasizes these differences well enough.

The authors also made substantial improvements in data presentation, addressing all my previous concerns. The data is now much clearer, and improved data presentation for validation of their RFT model (Figure 2e).

The authors also performed additional experiments to address the origins of bimodality in the expression of flagellar genes. They now argue that both YdiV and FlgM contribute to that bimodality.

Finally, they also tested the effect of balanced exponential growth on flagellar gene expression. Instead of using cells recovering from saturated stationary-phase cultures, they seeded their experimental cultures with "pre-cultures" that had not reached saturation. This approach mirrors the method used in Honda 2022. Importantly, even under these conditions, their observation that swimming velocity plateaus at WT levels of flagellar gene expression remains true, further supporting their findings.

With these improvements, the unique contributions of this study have been made much clearer. My take is that whereas the Honda et al. (2022) study demonstrated that bacteria regulate flagellar expression in a manner that keeps the flagellar number constant, the argument for why that specific flagellar expression level is a good choice, namely that it is the 'minimum necessary', remained rather conjectural. The present study makes much more concrete what is meant by the 'minimum necessary' investment - it is the minimum expression level of the regulon needed to achieve the maximum average swimming velocity, and provides a clear demonstration experimentally (by substantially overexpressing the regulon) and theoretically (via RFT calculations; despite its caveats raised by reviewers 5&6, the authors now do a commendable job of demonstrating robustness of the conclusions they draw under different assumptions) that wildtype levels of regulon expression does indeed appear close to the minimum required for maximizing average swimming velocity. Moreover, by demonstrating this using a strain (MG1655) in which motility gene expression has not been artificially engineered, as well as wild isolate strains (ECOR collection), the generality of the argument is strongly enhanced.

Given all of the above, I feel that the manuscript is now much stronger, and I would recommend it for publication.

Reviewer #2 (Remarks to the Author):

We are thankful to the Reviewers #1 and #2 for their highly positive and detailed feedback on the revised version of our manuscript and for their valuable comments and suggestions on improvement of its previous version.

Reviewer #3 (Remarks to the Author):

The authors addressed my questions. I supported the publication of the manuscript.

We thank the Reviewer #3 for helping us making our paper clearer and more complete.

Reviewer #5 (Remarks to the Author)

The two main points we felt needed to be addressed for manuscript to be of sufficient general interest for Nat. Comm, have now been addressed in the revised manuscript.

The first one was on the difference between the Honda et al results and their observations, specifically that their observations hold true for a range of growth conditions, including steady-state exponential growth. Although the authors did not repeat the experiment in the NCM strain used by Honda et al, they show in the revised manuscript that their conclusions hold across a range of growth conditions, including steady-state growth. We think that this shows the results are general enough for wider audience. The authors also discuss the differences between their results and that of Honda et al, specifying that Honda and co-authors did not go above WT level (in fact it seems they could not go above it). While they attribute this to the difference in strains, specifically that NCM3722 was made motile by inserting an IS in front of the flhDC, which is true, we that MG1655 also has non motile WT isolates. Then, the growth rates we observe between these strains (NCM3722 nonmotile and MG1655 motile isolate) are not as big as authors seem to indicate in the response. Certainly, NCM3722 is faster, but not by much. We do however see differences in motile fraction between the two strains, specially when grown on glucose (we describe this further below). This is why we feel that rather than saying MG1655 or NCM3722 are more suited for studying and understanding swimming, a better conclusion might be that how high the expression can go in a given strain, and why, is an interesting question that is yet to be understood.

We thank the Reviewer #5 for the thorough feedback on the revised version of our manuscript. We are glad to see that we could address both major points previously raised by this Reviewer. We also thank the Reviewer for additional suggestions regarding discussion and data presentation, which we have now addressed in the final revision of the manuscript.

*Although non-motile variants of MG1655 indeed exist, our point is that the canonical MG1655 strain used in our study is naturally motile, meaning that its motility was shaped by the natural evolutionary selection. This is different from motility of the NCM3722 strain that was artificially activated and thus never exposed to the evolutionary tuning (not even under laboratory conditions). We believe that this is a principal difference, and would argue that MG1655 is indeed better suited for motility analyses. Furthermore, regarding different growth rates between both strains, Cremer et al., 2019 (Table S2) reported the difference between 0.92 of NCM3722 and 0.72 of MG1655 (for growth on glucose), which is very substantial and clearly demonstrates that these strains differ in their physiology. Since both laboratory strains and natural isolates cover a broad range of motile abilities, we agree that there cannot be only one benchmark strain for studying motility. Yet our comparison of these different isolates strongly suggests that the expression-motility relationship is conserved among them and that MG1655 is a representative *E. coli* strain with high motility. We now mention this specifically in the text.*

We also pointed to the RFT model and how it was used compared to previous results. What the authors have presented now is clear, including how/why they picked coefficients different from those of eg Lighthill or Gray & Hancock. This has now been fully clarified and carefully investigated and we believe will be a service to the community. However, the authors might have misinterpreted our comment about the torque under which motors operate (page 16 of the letter). We do not have any issue with the results and overall interpretation that physics limits the maximum swimming speed, as opposed to by under-investment in flagella production. We also agree that motors operate in the low torque regime in those experiments, but we do not think the model can properly estimate the torque actually delivered by individual motors in bundles, because it neglects all filament-filament and filament-body interactions. This is why we suggested removing Extended Fig. 5g, because we are not sure those values actually correspond to the effective torque delivered by the motors, until estimates of internal friction can be obtained. There is a risk that presenting values without experimental backing or a more complete model will ultimately lead to confusion, similarly to the opposite belief that motors of swimming cell operate in the high torque regime, a view that has been prevalent in the literature (especially in theoretical studies) until fairly recently. We think this should be corrected accordingly.

We thank the Reviewer for raising this point and agree that calling the quantity we plot “motor torque” might get over-interpreted and we thus renamed the quantity as “bundle torque / N” and we modified the main text (L225):

*“The constant motor speed further **implies in our model** that the load per motor is low **even for single flagellated cells** and decreases as the number of flagella increases (Supplementary Fig. 5i), because multiple motors share the torque generation necessary for bundle rotation and cell propulsion. **Although the model neglects possible additional torque resulting from solid friction between flagella**, this result strongly suggests that under our conditions the motors operate in the low-torque regime at near-maximum speed, reported to vary from 100 to 300 Hz dependent on temperature.”*

Supplementary note 2, p.5:

*“The predicted torque generated by each motor thus decreases as a function of N (Extended Data Fig. 5i), and never exceeds 700 pN.nm, far from the ~1000-2000 pN.nm maximal torque that the motor can generate. **Although the model neglects solid friction between filaments**, these*

torque values imply that the motor operates close to or at the maximum speed limit of the torque-speed relation, which is consistent with our modeling hypothesis and our observations of the motor speed being constant, presumably maximal, for all strains”

And Supplementary note 2, p.10:

*“However, we cannot rule out **that solid friction between flagella increases the torque per motor to the level of a single flagellum** or that additional mechanisms regulate the motor speed in swimming cells somewhat below this maximum, e.g. via the dynamics of stator units or such mechanisms as *ycgR*-based regulation²⁴.”*

*To mention the possible effect of solid friction between flagella. We still believe that it is important to show this quantity in the supplementary information as a check of self-consistency of our model. Furthermore, we now additionally refer to Niu et al., *Sci Adv* (2023) (new ref. 52), which further supports the scenario of swimming at low torque and quasi-maximal speed in water, together with refs. 50 and 51, and we apologize for having missed this reference in the previous version.*

We have a few minor points for authors to consider:

- No values are given for growth rate despite looking specifically at the influence of different growth conditions (and mentioning this in the response to reviewers as an important difference). The competition assays are indeed a better indication of the fitness cost/advantage, but it would still be useful to include some numbers for batch/steady-state cultures and discuss/summarize them. It seems that the authors have measured growth curves for most conditions so this should be straightforward.

We thank the Reviewer for the suggestion, yet since the effect of different carbon sources on growth rates and motility genes expression was already investigated in our previous work (Ni et al., 2020), we did not quantify growth rates systematically and with sufficient number of replicates as it was not the central point of the current study. That is why we cannot provide reliable specific numbers, besides showing examples of the growth curves as is done already in Supplementary Fig. 1 and 10. But as the Reviewer also agrees, these values are not essential for our conclusions, and we now once more refer to the carbon-source dependence of the growth rate observed in Ni et al., 2020 when introducing the carbon-limited growth.

- We also struggled to get good motile fractions when growing MG1655 WT in glucose media, while NCM3722 maintains good motility in similar conditions. But in Extended Data Fig. 2: Swimming characteristics in liquid media (well-mixed conditions, no gradients). | Nature, Cramer et al show that MG1655 grown with glucose (triangle at 0.8 h⁻¹ growth rate) has the same speed as in other media. The authors should show the speed and motile fraction separately, so that one can see why the average speed is so low in the current study. It could be because the motile fraction is low while the speed of motile cells is still high?

We thank the Reviewer for this suggestion. We now present these data in the Supplementary Fig. 6. Both the fraction of motile cells and their swimming speed contribute to the decrease in the average swimming velocity.

- L22-23: low average expression levels?

We thank the Reviewer for the careful reading of our manuscript and pointing out this typo. The sentence was corrected.

- L48-49: now referring to papers that reported trade-off during experimental evolution, but with same reference list, so not sure it addresses the comment. No evolution in Honda et al?

We apologize for this mistake and thank the Reviewer for noticing it. There was indeed no experimental evolution in Honda et al, so this reference was removed at this place.

- L66: we think the present study does not determine those factors either but is showing that cells in rich media are close to the optimum spot on the investment/motility landscape.

We believe the Reviewer might have slightly overinterpreted our sentence from the Introduction (“In contrast, the factors that determine the upper physiological limit of resource allocation in motility remained unknown”). By saying this, we are only justifying the motivation of the present study without saying that now we have revealed all factors. However, we do believe that our data on flagella labeling combined with the model predictions unambiguously indicate that hydrodynamics of flagellar propulsion is one of the key factors defining the general upper limit of resource allocation in motility. But we are not claiming it to be the only factor.

- L133-134: data for swimming speed and motile fraction? Would include it in Extended Figure 1 to show why RP437 performs less well than MG1655: lower motile fraction or lower swimming speed or both?

We thank the Reviewer for this suggestion. In the case of RP437, the lower swimming velocity explains its worse swimming. Now the corresponding data for all E. coli K-12 strains were added to the Supplementary Fig.1.

- L138-142: the RpoS deletion experiment and the fact that the authors see no strong difference with the wild type in Fig 1d is interesting. But they only do it in TB, where the expression level of the wild type is already high, and motility near optimal. I don't think this experiment is sufficient to conclude that RpoS doesn't explain the difference between the strains used in this work and the NCM strain used by Honda et al, even if it may very well be true. A more meaningful experiment would be to repeat the experiment in minimal media with the RpoS mutant: this is in those conditions that NCM3722 and MG1655 differ the most. While RpoS levels are indeed low in exponential phase, it seems that E. coli K12 RpoS mutants show higher expression of some motility (MotA/B) and chemotaxis genes during exponential growth, eg Table 2 in <https://link.springer.com/article/10.1007/s00438-007-0311-4#Sec2> (same authors as ref 34-35)

While such comparison between MG1655 WT and MG1655ΔrpoS in minimal medium might be potentially relevant for further investigation of the differences between NCM3722 and MG1655, we would like to point out that this is not the objective of our study. We believe that the key discrepancies between our study and Honda et al. have been already clarified by the experiments

we already performed for the previous revision (which were, undoubtedly, important and we thank the Reviewer for proposing them), and any further investigation of the differences between the physiology of these strains goes beyond the scope of this study.